# Natural Products as New Approaches for Treating Bladder Cancer: From Traditional Medicine to Novel Drug Discovery

**DOI:** 10.3390/pharmaceutics15041117

**Published:** 2023-03-31

**Authors:** Yoo Kang, Chelin Park, Heemin Lee, Sojin Kang, Chunhoo Cheon, Bonglee Kim

**Affiliations:** 1College of Korean Medicine, Kyung Hee University, Seoul 02447, Republic of Korea; 2Department of Pathology, College of Korean Medicine, Kyung Hee University, Seoul 02447, Republic of Korea; 3Korean Medicine-Based Drug Repositioning Cancer Research Center, College of Korean Medicine, Kyung Hee University, Seoul 02447, Republic of Korea

**Keywords:** angiogenesis, bladder cancer, clinical trials, metastasis, natural product, resistance, traditional medicine

## Abstract

Bladder cancer (BC) is a heterogeneous disease that a tumor develops in the bladder lining and in some cases, the bladder muscle. Chemotherapy and immunotherapy are commonly used to treat bladder cancer. However, chemotherapy can cause burning and irritation in the bladder while BCG immunotherapy, which is the main type of intravesical immunotherapy for bladder cancer, can also cause burning in the bladder and flu-like symptoms. Thus, drugs originating from natural products have attracted much attention due to the reports that they have anti-cancer properties with low adverse effects. In this study, eighty-seven papers that dealt with natural products preventing or treating bladder cancer were reviewed. The studies were classified into the following mechanism: 71 papers on cell death, 5 papers on anti-metastasis, 3 papers on anti-angiogenesis, 1 paper on anti-resistance, and 7 papers on clinical trials. Most of the natural products that induced apoptosis up-regulated proteins such as caspase-3 and caspase-9. Regarding anti-metastasis, MMP-2 and MMP-9 are regulated frequently. Regarding anti-angiogenesis, HIF-1α and VEGF-A are down-regulated frequently. Nevertheless, the number of papers regarding anti-resistance and clinical trial are too few, so more studies are needed. In conclusion, this database will be useful for future in vivo studies of the anti-bladder cancer effect of natural products, in the process of selecting materials used for the experiment.

## 1. Bladder Cancer

Bladder cancer (BC), having gone through various natural historical events, differentiates from other diseases [1]. It is a tumor developed in the bladder lining as tissues have grown abnormally, and in some cases, even spread into the bladder muscle directly [2]. Blood in the urine appears most commonly, being the second most common genitourinary malignancy, although it is painless and has increased in the number of global survivors [2]. Global cancer statistics 2020 has shown that BC is the 10th most diagnosed cancer worldwide, with about 573,000 new cases and 213,000 deaths [3]. It is more common in males, with respective incidence and mortality rates of 9.5. In addition, 3.3 per 100,000 among men has passed away, which is approximately 4 times than those among women [3]. As awareness toward BC has grown among global citizens over the past two decades, the quality of life ramifications of BC diagnosis, treatment, and surveillance have also increased on top of the oncologic perspective [2]. 

## 2. Diagnosis

When diagnosed, most BC is urothelial, being non-muscle-invasive (75%), with half of it being low-grade tumors. The concept of “non-muscle-invasive” is opposed to “muscle-invasive or disseminated (25%)”, which is mostly high-grade. The most important prognostic factor is cellular grade, which in the case of BC had been divided into stages according to the T scale: Ta, T1, T2, T3, or T4. Here, T2 is the stage which one reaches in the case of a muscle-invasive tumor [4]. 

As implied, the nomenclature currently divides BC into non-muscle-invasive or musc le-invasive. Such way of defining the standards suggests a large difference in the perspective of therapeutic options and vital prognosis. Non-muscle-invasive tumors that have been classified as T1 (involvement of the lamina propria) are unique that they refer to the invasive nature of BC. Nevertheless, other various data should also be considered when making therapeutic decisions as the inter-observer variability is important [4].

## 3. Treatment: Present and Future

BC is high in incidence, disease, and death rate. Since the clinical initiation, primeval diagnostic approval and multi-treatment include various medical specialties. Late approaches related with immunology and chemotherapy make it essential to refer to future aspects [4].

Growing older, male sex, and smoking increase the rate of BC. BC results in tumors that turn out with gross or microscopic hematuria, evaluated with cystoscopy and the imaging of the upper tract. Treatments such as endoscopic resection and adjuvant intravesical therapy are proceeded to cure non-muscle-invasive tumors, according to the risk classification. Technology to detect tumors well and decrease the rate of recurrence is included in intensified cystoscopy. Although Bacille Calmette-Guérin (BCG) is the standard-of-care immunotherapy, there still are patients who do not respond to adjuvant therapy. For such “challenging” patient populations, many other treatments are being studied to manage them [5].

Those going through muscle-invasive disease are warranted to curb the risk of metastasis and disease-specific mortality through other therapies. For instance, urinary diversion or trimodal therapy and more aggressive therapy with radical cystectomy are adapted. Changes are being made rapidly among the advanced disease therapies as immunotherapy has become an option for certain patients with various stages of disease [5].

Chemotherapy, however, can cause burning and irritate the bladder. Such burning can also occur with BCG immunotherapy, which is the main intravesical immunotherapy for BC [6]. The main side effect of BCG is flu-like symptoms, including fever, chills, and fatigue. Although there are a number of drugs and combinations of drugs, few common chemo side effects are known [7]. It includes changes in appetite, diarrhea, fatigue, fever, infection, hair loss, memory problems, mouth and throat sores, and nausea and vomiting. Meanwhile, radiation therapy destroys cancer cells by using high-energy X-rays or particles [8].

Various living organisms act as sources for natural products [9]. Bioactive compounds have the potential to prevent or treat the major diseases and thus have been used as therapeutic medicine in human history [9]. Lately, they have continuously provided a key basis for the development of drugs including *Nigella sativa* L. [10], *Salvia miltiorrhiza* [11], *Spatholobus suberectus* Dunn [12], and *Morus alba* L. [13]. Like such, bioactive products are known to show biological actions, including anti-cancer activities. Studies have given more attention recently to naturally derived products and their role of decreasing oxidative stress caused by free radicals [9]. Thus, it is inevitable and essential to develop novel drugs to treat bladder cancer with natural products and herbal medicine.

In this current study, 87 papers about natural products used to prevent or treat BC were reviewed. They were classified according to the mechanism of bladder cancer—71 papers on cell death, 5 papers on anti-metastasis, 3 papers on anti-angiogenesis, 1 paper on anti-resistance, and 7 papers on clinical trials—will help medical staff treat it and contribute to society. The main acting mechanism was “cell death” as cancer is a disease in which some of the body’s cells grow uncontrollably and spread to other parts of the body.

## 4. Natural Products and Cell Death

We reviewed 87 papers written about natural products used to prevent or treat bladder cancer. The main acting mechanism “cell death” is the result of several distinct processes, which are apoptosis, necrosis, pyroptosis, oncosis, and autophagy [14]. The most well-known processes are apoptosis and necrosis, in which cells are removed in a controlled and uncontrolled manner, respectively [14]. Fifty-five papers were dealing with phytochemicals that induced cell death, and the papers that dealt with the efficacy of phytochemicals on bladder cancer were written mainly on “induction of apoptosis” along with induction of cell cycle arrest, autophagy, uro- and nephrotoxicities, cytotoxicity, and genotoxicity together with an inhibition of oxidation, proliferation, and invasiveness. Most of the experiments changed the dose of a specific chemical while maintaining the same duration or vice versa, while the rest focused on the IC_50_ value, in which 50% of the cells die (apoptosis).

### 4.1. Single Compounds and Apoptosis

Forty-three papers were written about the apoptotic effects of single compounds (Table 1).

You et al. tested the cytotoxicity of compounds isolated from *Murraya tetramera* on EJ bladder cancer cells [15]. Compound **4** (5,7-dimethoxy-8-[(Z)-3′-methyl-butan-1′,3′-dienyl]coumarin) exhibited potent cytotoxicity against EJ, with IC_50_ values of 30.59 µg/mL.

Blazevic et al. tested the cytotoxicity of hydrodistillate and allyl isothiocyanate (originated from *Lepidium latifolium* L.) on human bladder cancer cell UM-UC-3 [16]. Hydrodistillate, and allyl isothiocyanate inhibited UM-UC-3 with the IC_50_ value of 192.9 μg/mL, and 23.27 μg/mL for 24 h, respectively. All these methods can be considered as moderately active.

Tsai et al. revealed that the mechanism of miR99a expression occurred by BITC treatment on T24 and 5637 bladder cancer cells [17]. Treatment of 20 µM BITC for 24 h on T24 and 5637 cells dramatically improved JNK and ERK activation, suggesting that JNK and ERK may contribute to miR-99a expression. Moreover, by pre-treating 10 µM of U0126 (ERK inhibitor) for 30 min on T24 and 5637 cells, ERK activation as well as miR99a expression diminished, while inhibition of JNK activation showed a nonsignificant response.

Gerhardt et al. reported that boldine isolated from *Peumus boldus* has therapeutic effect on bladder cancer, inducing apoptosis in T24 cells [18]. Cells were treated with 200, 300, 400, 500 μM of boldine for 24 h. Cells treated with boldine showed decreased phosphorylation of ERK, and cells treated especially with 400 μM of boldine showed a decrease in AKT and GSK-3β phosphorylation. These changes induced apoptosis by the activation of GSK-3β and inactivation of ERK, AKT.

Cao et al. revealed that cordycepin from *Cordyceps militaris* (*C. militaris*) hot water extracts induced apoptosis in T24 human bladder cancer cells [19]. Cordycepin-CMHW was given to cultured T24 cells. T24 cell survival decreased in a dose-dependent manner, which was seemingly mediated by activation of A3AR and Caspase-3 along with suppression of Akt, leading to the inactivation of Lef/Tcf, β- catenin-responsive C-myc and Cyclin D1. These alterations of proteins and genes expression led to the induction of apoptosis.

Phenolic alkaloids of Menispermum dauricum (PAMD), extracted from *Menispermum dauricum*, mainly consist of dauricine [20]. Using MTT assay, concentration above 8 μg/mL to 64 μg/mL for 24–72 h of dauricine showed an inhibitory effect on EJ cells. There was a correlation between the drug concentration and the inhibition rate but no relationship between the extension of time and the inhibition rate.

Cha et al. demonstrated that Emodin modifies two epigenetic markers to suppress cancer inflammation-associated gene expression [21]. Emodin decreased the level of pH3Ser10 and increased the level of H3K27me3 in MBT2, T24, TSGH8301, and J82 cells. This phenomenon was more prominent in MBT2 and T24 than in TSGH8301 and J82. Emodin-treated T24 cells showed attenuated HBP17 and FABP4. The CHIP assay results of those cells confirmed decreased pH3Ser10 and increased H3K27me3 markers on both HBP17 and FABP4 promoters, and the binding of RNA polymerase II was dramatically decreased.

Abbaoui et al. reported that broccoli and broccoli sprout isothiocyanates, especially sulforaphane and erucin, have efficacy of bladder cancer growth inhibition both in vitro and in vivo [22]. Tumor growth rates in UMUC3-bearing female athymic mice showed an 82% inhibition by erucin treatment at a dose of 295 μmol/kg/day for 2 weeks with decreased Ki67-positive cells, which are markers of cell proliferation.

Matin et al. found that ferutinin extracted from *Ferula ovina* has a pro-apoptotic effect on TCC bladder cancer cells [23]. The IC_50_ values of ferutinin for 72 h on TCC cells were measured at 24 µg/mL, which was a higher toxicity compared with vincristine (an anti-cancer drug used for bladder cancer treatment). In addition, treatment with IC_50_ value of ferutinin for 24 h (33 µg/mL) reduced the number of cells, their attachment, and also induced cytoplasmic granulation.

Amigo-Benavent et al. revealed that formononetin from *Astragalus membranaceus* (*A. membranaceus*) inhibited human bladder cancer cell T24 proliferation and invasiveness via regulation of miR-21 and PTEN [24]. Cells were treated with 0, 50, 100, and 200 μM/mL of formononetin for 48 h. As a result, PTEN increased while miR-21 and p-Akt decreased in formononetin-treated T24 cells. These alterations of proteins and genes expression led to the induction of apoptosis.

Han et al. elucidated that Fucoidan extracted from *Fucus vesiculosus* (*F. vesiculosus*)-induced ROS dependent apoptosis in 5637 human bladder cancer cells [25]. Cells went through two independent processes. They were first exposed to *Fucoidan* at durations of 0, 6, 12, 24, 48 h with a dose of 100 μg/mL while in the other they were treated with Fucoidan at doses of 0, 10, 25, 50, 100 μg/mL for 24 h. It after all led to an increase in the Bax/Bcl-2 ratio and generation of the intracellular ROS. These alterations of proteins and genes expression resulted the induction of apoptosis.

Wang et al. reported that fucoxanthin extracted from *Undaria pinnatifida* plant induced apoptosis in T24 bladder cancer cells [26]. Cells were treated with 20 and 40 μM of fucoxanthin for 48 h. Treatment of fucoxanthin up-regulated mutant-type p53, caspase-3, and down-regulated mortalin-p53. These alterations of proteins and genes expression led to the induction of apoptosis.

Olarte et al. tested the anti-tumor properties of Hexane fraction and TLC-pure f61 extracted from *Cassia alata* L. [27]. Hexane fraction showed cytotoxicity to T24 at concentrations of 100, 50, and 25 µg/ mL. TLC-pure f6l, isolated from active Hexane fraction (FB) demonstrated cytotoxicity to T24 with IC50 values of 17.13 µg/mL. Moreover, after exposure of 50 µg/mL f61 for 12–24 h, T24 cells rounded up and lost contact with neighboring cells and the substratum of the wells.

Wu et al. reported the anti-proliferative effect and the apoptosis mechanism of kaempferol on EJ bladder cancer cells [28]. EJ cells were treated with 20, 40, 80 μM of Kaempferol. Exposed cells showed an up-regulated expression level of p-p53. Expression of p-p53 affected the mitochondria-mediated apoptotic signaling pathways, characterized by activated Bax, Bad, and inactivated Bid, Mcl-1, Bcl-xL. Furthermore, they showed a decreased level of p-AKT, which regulates cell progression and inhibits pro-apoptotic protein expression. These sequential changes of protein expression induced apoptosis of EJ cell.

Park et al. showed that *Broussonetia papyrifera* (*B. papyrifera*) induced cytotoxic effects against the human bladder cancer cells, T24 and T24R2 [29]. Six different human cell lines were treated with 20 μM of compound **1** (flavonol). As a result, the fold (gene/β-actin) decreased in the case of Cyclin D1 while increasing in p21′s case. Thus, inhibition of tumor cell growth may be the result of cell cycle arrest at G0/1, induced by such changes of protein levels.

Kuan-Sin-Yin (KSY) decoction is a popular traditional Chinese herbal medicine that had been used to reinforce the qi [30]. The HPLC analysis showed that 1 g KSY contained 29.59 ± 0.78 μg of calycosin-7-*O*-β-d-glucoside, 187.08 ± 4.28 μg of liquiritin apioside, 228.42 ± 2.91 μg of liquiritin, and 701.19 ± 7.92 μg of glycyrrhizic acid. The tumor volumes of the KSY-treated mice were reduced in the MBT-2 bearing C3H/HeN, BALB/cAnN-Foxn1nu mice. The expression of cell cycle regulators, p53 and p21 in KSY-treated BALB/cAnN-Foxn1nu mice, were increased. In vitro, the proliferation of MBT-2 cells was suppressed by KSY treatment.

Licochalcone A (LCA) is a flavonoid extracted from root of licorice, the most commonly prescribed herbs in China for various diseases [31]. The effect of LCA-induced proliferation inhibition on T24 cells was evaluated by sulforhodamine B (SRB) assay, and the IC50 was approximately 55 µM. Moreover, LCA treatment enhanced the ROS level and decreased the GSH/GSSG ratio in T24 cells. Therefore, the study determined that LCA suppresses proliferation of T24 cells by inducing oxidative stress responses.

Lida et al. found that Luteolin has an apoptotic effect and inhibitory effect on T24 and 5637 human bladder cancer cells [32]. T24 cells treated with 10, 25 µmol/L of Luteolin for 48 h and 5637 cells treated with 25 µmol/L of Luteolin for 48 h showed apoptosis. Treatment of Luteolin on T24 cells up-regulated Cdc25c, Cdk2, p21^Waf1/Cip1^, p27^Kip1^, p53 and down-regulated Cdk4, cyclin A, cyclin D1, p-S6, p-p70S6K, ROS. Treatment of Luteolin on 5637 cells up-regulated p21^Waf1/Cip1^ and down-regulated p-S6.

Chiu et al. found that *Radix Angelica Sinensis* (RAS) induced apoptosis in BFTC human bladder cancer cells [33]. BFTC human bladder cancer cells were treated with 60 μg/mL of N-butylidenephthalide (BP) for 72 h. BP induced apoptosis by activating caspase-9, caspase-3, E-cadherin, and down-regulating N-cadherin.

Kaiser et al. evaluated the cytotoxicity of oxindole alkaloid purified fraction (OAPF) (originated from *Uncaria tomentosa*) and its isomerized form on T24 and RT4 cancer cells [34]. After heating OAPF under reflux at 85 °C for 5, 15, 45 min, the alkaloids were isomerized. OAPF, OAPF 5′, OAPF 15′, and OAPF 45′ were treated on T24 and RT4 cells for 48 h.

*Rhodiola rosea* L., which consists of salidroside, is a perennial herbaceous plant of the Crassulaceae family [35]. Treatment of salidroside for 8 h increased the presence of LC3-GFP puncta in UMUC-3/pEGFP-LC3 cells by 64%, whereas the control treatment only showed about 8% cells with LC3-GFP puncta. Furthermore, Western blotting analysis revealed that salidroside treatments degraded p62.

Backer et al. reported that Triterpene glycoside extracted from *Pittosporum angustifolium* plant has an anti-proliferative effect on 5637 bladder cancer cells [36]. The IC_50_ values of Saponin **1**, **2**, **3**, **4**, and **10** were measured 4.1, 5.2, 2.1, 17.9, and 2.4 µM, respectively. All the five compounds displayed the same acylation pattern with angelic acid at C-22 of the A1-barrigenol aglycone. In addition, the relatively weaker anti-proliferative effect of Saponin **4** can indicate that sugar units affect cytotoxicity, since glycoside 4 possesses a differently composed trisaccharide chain than compounds **1**–**3** and **10**, and the aglycone part is totally identical.

Tapondjou et al. reported the cytotoxicity of steroidal saponins extracted from the flowers of *Dioscorea bulbifera var. sativa* on ECV-304 bladder cancer cells [37]. Among the 15 isolated saponin compounds, compound **12** (pennogenin 3-*O*-α-l-rhamnopyranosyl-(1→4)-α-l-rhamnopyranosyl-(1→4)-[α-l-rhamnopyranosyl-(1→2)]-β-d-glucopyranoside), compound **13** (26-*O*-ß-d-glucopyranosyl-(25R)-5-en-furost-3ß,17α,22α,26-tetraol-3-*O*-α-l-rhamnopyranosyl-(1→4)-α-l-rhamnopyranosyl-(1→4)-[α-l-rhamnopyranosyl-(1→2)]-β-d-glucopyranoside) and compound **15** (spiroconazol A) showed a cytotoxicity with IC_50_ values of 8.5, 14.3, and 5.8 μg/mL, respectively.

Wang et al. reported that sulforaphane existing exclusively in cruciferous vegetables suppresses cancer growth [38]. Athymic mice were injected subcutaneously with a UM-UC-3 cell and sulforaphane from broccoli sprout was treated at doses of 12 mg/kg body weight for 5 weeks. The average tumor volume decreased by the inhibitory rate of 63%, caspase 3 and cytochrome c expression were induced, and the expression of survivin was reduced in the sulforaphane treated mice.

Abbaoui et al. reported the anti-cancer effect of sulforaphane in bladder cancer [22]. Tumor growth rates in UMUC3 bearing female athymic mice showed a 42% inhibition by sulforaphane treatment.

Chang et al. evaluated the cytotoxicity of Tanshinone IIA Nanoemulsions (TA-NEs), originating from *Salvia miltiorrhiza* bunge [39]. The treatment of TA-NE-F4 on T24 cells caused 50% cell death at concentrations of 13.48 ± 4.30 μg/mL for 24 h. Moreover, the treatment of TA-NE-F4 on T24 cells for 24 h caused cell death, based on the decreased cell number and shrinkage in morphology.

Cao et al. elucidated that Tea polyphenols inhibited autophagy and induced apoptosis in T24, BIU87 bladder cancer cells [40]. The IC_50_ of EPI and TP were 8.6 and 399.6 μM in T24 cells while 38.3 and 670.8 μM in BIU87 cells. As a result, LC3-II and JNK were significantly increased, while p62 and Bcl-2 expression decreased in a time-dependent manner. These alterations of proteins and genes expression led to the inhibition of autophagy and the induction of apoptosis.

Olarte et al. tested the antitumor properties of Hexane fraction and TLC-pure f61 extracted from *Cassia alata* L. [27]. Hexane fraction and TLC-pure f6l, isolated from active Hexane fraction (FB), both demonstrated cytotoxicity to T24. After exposure of f61, T24 cells rounded up and lost contact with neighboring cells and the substratum of the wells. They also exhibited chromatin condensation, formation of membrane blebs, apoptotic bodies, and reduction in overall size, which are the morphological features associated with apoptosis.

Mu et al. elucidated the structure of new triterpenoid saponin compound (**1**, **2**) and reported the cytotoxicity of Triterpenoid saponin (**1**, **2**, **4**, **5**) extracted from *Ardisia gigantifolia* on EJ bladder cancer cells [41]. Triterpenoid saponin **1**, **2**, **4**, **5** showed the IC_50_ value of 3.4 ± 0.1 µM, 4.0 ± 0.4 µM, IC_50_ 2.0 ± 0.1 µM, 3.1 ± 0.2 µM for 48 h, respectively.

Zhang et al. reported activity of Yuanhuacine (YHL-14) against bladder cancer cell proliferation and its mechanism [42]. Cell growth rate was inhibited significantly in YHL-14-treated two bladder cancer cell lines, T24T, UMUC3. YHL-14 treatment at a dose of 2 µM for 12 h activated p38 in T24T, which leads to Sp1 protein accumulation and transactivation, subsequently resulting in p21 gene transcription and protein expression.

You et al. tested the cytotoxicity of compounds isolated from *Murraya tetramera* on EJ bladder cancer cells [15]. Compound **1** (β-eudesmol) showed strong cytotoxic activity against EJ, with IC_50_ values of 31.93 µg/mL.

Cao et al. did an experiment with cordycepin from *Cordyceps militaris* hot water extracts [19]. However, the duration for which the treatment was applied did not seem to be listed on the paper, whereas most of the other papers had clearly written the dose and duration of the application. The chemical structures of the compounds were presented (Figure 1) [43].

### 4.2. Single Extracts and Apoptosis

Twenty-five papers were reported about the apoptotic effects of single extracts (Table 2).

Rafieian-Kopaei et al. revealed that *Juniperus foetidissima* (*J. foetidissima*) induced cytotoxic effects against the EJ-138 bladder and CAOV-4 ovary cancer cells [44]. The compounds extracted from *J. foetidissima* exhibited cytotoxic effects in a dose-dependent manner with values of 43.26 ± 3.21, 44.27 ± 4.25, and 26.17 ± 2.96 μM against the EJ-138 cell line, and values of 25.72 ± 3.13, 57.38 ± 4.56, and 37.35 ± 4.13 μM against the CAOV-4 cell line, respectively. Such alterations led to the induction of cytotoxicity.

Bilusic et al. tested the pro-apoptotic effect of wild asparagus (*Asparagus acutifolius* L.), black bryony (*Tamus communis* L.), and butcher’s broom (*Ruscus aculeatus* L.) aqueous extracts on T24 cells [45]. The highest increase in late apoptosis was achieved by 1 mg/mL of T. communis aqueous extracts.

Begnini et al. tested the apoptotic effect of Brazilian red propolis (BRP) ethanolic extract on 5637 bladder cancer cells [46]. The concentration of 50 μg/mL was effective in inducing early apoptosis, while the concentration of 100 μg/mL was effective in both early apoptosis and late apoptosis/death. A total of 50 μg/mL of BRP ethanolic extract up-regulated Bcl-2, Bax, AIF, caspase-9, caspase-3, p53 and down-regulated the Bax/Bcl-2 ratio. However, 100 μg/mL of BRP ethanolic extract altered the regulation in a contrasting way, which was the up-regulation of the /Bcl-2 ratio and the down-regulation of Bax, Bcl-2, caspase-3, caspase-8, caspase-9.

Lou et al. reported that *Brucea javanica* (*B. javanica*) oil induced apoptosis in T24 bladder cancer cell [47]. Cells were treated with *B. javanica* oil at doses of 0.078, 0.156, 0.313, 0.625, 1.25, 2.5, and 5 mg/mL for 48 h. *B. javanica* oil treatment down-regulated NF-κB, COX-2 expression while up-regulating the expression of caspase-3, -9. These alterations of protein and gene expression led to the induction of apoptosis.

Bilusic et al. tested the pro-apoptotic effect of wild asparagus (*Asparagus acutifolius* L.), black bryony (*Tamus communis* L.) and butcher’s broom (*Ruscus aculeatus* L.) aqueous extracts on T24 cells [45]. Treatment of butcher’s broom extract on T24 cells in a dosage of 0.5, 1, 2 mg/mL for 48 h induced apoptosis.

Chen et al. showed that triterpenoid erythrodiol extracted from *Celastrus kusanoi* (*C. kusanoi*) stems induced apoptosis in NTUB1 cells [48]. Cells were exposed to triterpenoid erythrodiol at doses of 5, 10 μM for 24 h. It significantly elevated the amount of ROS, leading to the cell cycle arrest at G0/G1 accompanied by an increase in the extent of apoptotic cell death. These alterations of proteins and genes expression led to the induction of apoptosis.

Blazevic et al. tested the cytotoxicity of dichloromethane leaf extract on human bladder cancer cell line, UM-UC-3 [16]. Dichloromethane leaf extract exerted cytotoxicity effect in UM-UC-3 with the IC_50_ value of 133.8 μg/mL for 24 h, respectively.

Ahn et al. revealed that ethanol extracts of *Citrus unshiu Markovich* (CUM) peel induced apoptosis in T24 human bladder cancer cells [49]. Treated with 0, 100, 200, 400, 600, 800, 1000 μg/mL of EECU for 48 h, the cell viability was measured by an MTT assay. EECU-induced apoptosis was found to correlate with an activation of caspase -8, -9, and -3, the generation of ROS while inactivating P13K, Akt, and LY294002. Such alterations led to the induction of apoptosis.

Kim et al. reported the bladder cancer prevention activity of garlic (*Allium sativum* L.) extract in the T24 cell xenograft model and its mechanisms [50]. Compared to the control group, significant differences in tumor volume and tumor weight were observed in garlic extract groups. AKAP12, RDX, and RAB13 were identified with associated genes with the PKA signaling pathway. Furthermore, AKAP12 and RDX were increased and RAB13 was decreased in garlic feeding groups, but the expression value of these genes in the data of 165 bladder cancer patients was reversed.

Miranda et al. tested the cytotoxicity of glycoalkaloidic extract (AE) extracted from *Solanum lycocarpum* and nanoparticles loading AE (NP-AE) on RT4 bladder cancer cells [51]. The IC_50_ values of NP-AE and free AE for 24 h were 4.18 μg/mL and 8.17 μg/mL, respectively. In an apoptosis assay, cells were treated with NP-AE and both NP-AE and AE significantly increased apoptotic cells. Furthermore, RT4 cells were cultured under 3D conditions, but the IC_50_ values were around three times higher in 3D spheroids compared to conventional monolayer cultures.

Raina et al. delineated the effect and mechanism of Grape seed extract (GSE) against bladder cancer cell lines, T24 and HTB9 [52]. While 25 μM of the GSE treatment had an apoptotic effect on HTB9 cells, in the T24 cells, the meaningful death was induced from 50 μM of GSE. Furthermore, the GSE treatment increased the expression of apoptosis-related molecules, cleaved caspase-3 and -9, cleaved-PARP, and decreased the expression of anti-apoptotic molecule Mcl-1 in both cells. The time-dependent increase in cleaved-PARP also indicated that apoptosis occurred around 6 h in the T24 cells, while in the HTB9 cells it occurred around 12 h.

Wu et al. elucidated that *Guizhi Fuling Wan* (GFW) induced cell cycle and apoptosis in BFTC 905 and TSGH 8301 bladder cancer cells [53]. GFW presented relatively high selectivity regarding cancer cells and minimal toxicity to normal urothelial cells and that it interferes with cell cycle progression through the activation of CHK2 and P21 and inhibits CDK2-cyclin E, A, and A complexes. These alterations of proteins and genes expression led to the induction of cell cycle and apoptosis in these bladder cancer cells.

Hamsa et al. showed that *Ipomoea obscura* L. (*I. obscura* L.) induced uro- and nephrotoxicities [54]. Swiss albino mice were treated with an acute dose of CP (1.5 mmol/kg body wt ip) and an alcoholic extract of *I. obscura* (10 mg/kg, body wt, ip) at durations of 4, 24, 48 h. It significantly elevated the level of IFN-γ and IL-2 while leading to the decrease in TNF-α. These alterations of proteins expression led to the induction of uro- and nephrotoxicities.

Bidinotto et al. found that Lemongrass essential oil (LGEO) extracted from *Cymbopogon citratus Stapf* attenuates cytotoxicity caused by carcinogen N-methyl-N-nitrosourea (MNU) on the urothelial epithelial cells of female BALB/c mice [55]. BALB/c mice were allocated into three groups. Group 2 was treated with the LGEO vehicle and MNU. Group 3 was also treated with LGEO and MNU but in a more frequent and higher dosage. As a result, Group 3 showed a significant reduction on cell proliferation and apoptotic indexes in urothelial epithelial cells compared with the group 2.

*Lycium barbarum polysaccharides* (LBP) is the major functional component of the fruit of *L. barbarum*, which is a well-known Chinese herb. P-AKT is characteristic of PI3K activation, closely correlated with cell proliferation and apoptosis [56]. The LBP treatment inhibited the proliferation of BIU87 cells and reduced the expression of p-AKT. The study demonstrated that LBP inhibits the proliferation BIU87 cells by repressing PI3K/AKT pathway.

Khan et al. elucidated that *Trillium govanianum* (*T. govanianum*) induced cytotoxicity [57]. Four human carcinoma cell lines (MCF7, HepG2, A549, EJ138) were treated with different concentrations of test samples (the MeOH extract and four SPE fractions). As a result, the four cell lines each showed = 5, 7, 9, 5 μg/mL, respectively. It is reasonable to state that *T. govanianum* could be exploited as a good source of cytotoxic compounds with putative anti-cancer potential.

Chen et al. elucidated that *Morus alba* (*M. alba*) increased the apoptotic effect of paclitaxel against the Aurora A and Plk1 cancer cells [58]. The combined treatment with 25, 500, 750, 1000, 1500 μg/mL of MWE (Mulberry Water Extract) had a duration of 24 h, and then extended to 48 h. As a result, up-regulation of p-cdc2, p-cyclin B1, p-aurora A, p-plk1, PTEN, caspase-3 occurred along with a down-regulation of EEA1. Such alterations led to an increase in the apoptotic effect of paclitaxel.

Kaiser et al. found that *Uncaria tomenotosa* (*U. tomenotosa*) induced both genotoxicity and cytotoxicity [59]. Human non-malignant cell line (human leukocytes) and human malignant cell lines (T24 and U-251-MG) were treated with *U. tomenotosa*. *U. tomenotosa* induced genotoxicity and cytotoxicity on T24 and U-251-MG. However, on human leukocytes, it showed different cytotoxicity.

Masci et al. reported that *Punica granatum* L. (*P. granatum* L.) had an antiproliferative effect on T24 cells [60]. Soxhlet extract from Isr peel showed a 54.3% inhibition of cell proliferation. Since ellagic acid was a predominant component of the mixture, a high correlation between its content and antiproliferative effect was shown.

Lee et al. showed that pomegranate fruit ethanol extract (PEE) from *Punica granatum* induced apoptosis in T24 cells [61]. Cells were exposed to 50, 100 μg/mL of PEE for 24, 48, 72 h. The PEE-treated cells showed the activated pro-caspase-3, -8, -9, -12, Bax/Bcl-2 ratio, CHOP, and Bip. These alterations of proteins and genes caused ER stress, mitochondrial damage, and death receptor signaling, which led to apoptosis.

Wu et al. reported the molecular pathway underlying the anti-cancer efficacy of Taiwan pomegranate fruit juice against urinary bladder urothelial carcinoma [62]. Pomegranate fruit ethanol extract (PFE) treatment inhibited T24 cell proliferation through restriction of the PTEN/AKT/mTORC1 pathway via profilin 1 up-regulation. It also evoked cell apoptosis through the over-expression of Diablo, which binds to XIAP and thus prevent XIAP to inhibit apoptosis.

Acai, which is a commonly consumed fruit, has pulp that includes the flavonoids anthocyanins and proanthocyanins, lignans, ascorbic acid, and others [63]. Intake of diet containing 5% AP for 10 weeks decreased the transitional cell carcinoma incidence and multiplicity, p63, and PCNA of male Swiss mice chemically induced to urothelial carcinogenesis for 10 weeks. Furthermore, acai fruit intake reduced the DNA damages induced by H2O2, presenting anti-carcinogenic activity against bladder cancer.

Hamsa et al. revealed that *Tinospora cordifolia* (*T. cordifolia*) induced apoptosis [64]. Swiss albino mice received 5 doses of *T. cordifolia* (200 mg/kg i.p.) and a single acute dose of Cyclophosphamide. It significantly elevated the level of GSH and IFN- γ, IL-2 while leading to the decrease in TNF-α. These alterations of antioxidant and proteins expression led to the induction of apoptosis.

Bilusic et al. tested the pro-apoptotic effect of wild asparagus (*Asparagus acutifolius* L.), black bryony (*Tamus communis* L.), and butcher’s broom (*Ruscus aculeatus* L.) aqueous extracts on T24 cells [45]. The highest increase in early apoptosis was achieved by 2 mg/mL of A. acutifolius aqueous extracts.

Hamsa et al. revealed how *Tinospora cordifolia* (*T. cordifolia*) induced apoptosis not only through the experiment itself, but by looking through the metabolisms that Cyclophosphamide (CP) undergoes in vivo [64]. The point here is that they thoroughly explained the role of the bladder as the primary storage organ for urine and how the sensitivity of the bladder to the damage induced by such metabolites is expected to be more.

Masci et al. reported that *Punica granatum* L. (*P. granatum* L.) induced the antiproliferative activity of T24 cells [60]. Along the way, they made a meaningful attempt of comparing their results on the total polyphenol content detected by the Folin–Ciocalteu assay with those already reported in the literature.

Chen et al. showed that triterpenoid erythrodiol extracted from *Celastrus kusanoi* (*C. kusanoi*) stems induced apoptosis in NTUB1 cells [48]. Although they did encourage the development of anti-cancer agents targeting G1 phase arrest, it would have been better if the detailed mechanism of 3-induced inhibition of tumor cell growth was elucidated.

### 4.3. Mixture Extracts and Apoptosis

Three studies reported about the apoptotic effects of mixture extracts (Table 3).

Radan et al. reported that *Coffea arabica* (*C. arabica*) and *Llex paraguariensis* (*L. paraguariensis*) inhibited the proliferation and induction of cytotoxicity in Caco-2, A549, OE-33, T24, CCD-18Co cells [65]. The green coffee bean extract (GCBE) and yerba mate extract (YME) were treated. As a result, caspases-8 and -3 were up-regulated while NF-kB, COX-2/PGE2, iNOS/NO, Topoisomerase II were down-regulated. These alterations of protein regulation led to inhibition of the proliferation and induction of cytotoxicity.

Gu et al. showed that *Centaurea ragusina* L. (*C. ragusina* L.) induced the cytotoxicity and apoptosis in T24, A1235 cells [66]. Treatments with flower and herba AE were applied in concentrations of 2 and 1 g/L, and were examined for the duration of 4, 24, 48, 72 h. As a result, inhibition of sulphydryl enzymes, phosphofructokinases, and glycogen activities occurred, which led to the induction of cytotoxicity and apoptosis.

Gong et al. tested the effect of Qici Sanling decoction (QCSL) on BALB/c-nu nude T24 xenograft mice [67]. Mice treated with QCSL showed an increase in survival time. Xenograft from QCSL-treated mice showed apoptosis. Tumor cells excised from QCSL-treated mice showed an inhibition of cell proliferation. Furthermore, the mechanism of antitumor effect of QCSL was revealed as an inhibition of the WNT/β-catenin pathway, by inactivating β-catenin, survivin, c-myc, and cyclin D1.

Gu et al. described specifically the chemical compositions of *Centaurea ragusina* L. aqueous extracts, which were quite unique compared to the others. Other studies usually focused on the experiment itself without analyzing the fractions of the extract [66]. The apoptotic mechanisms of the natural products were elucidated in Figure 2.

## 5. Natural Products and Metastasis

Metastasis is a major contributor to cancer-associated deaths [71]. We found three compounds and two single extracts reported to have anti-metastasis activity against bladder cancer cell (Table 4).

Chiu et al. tested the efficacy of N-butylidenephthalide (BP) both in vivo and in vitro [33]. In addition, they revealed its apoptotic effect and anti-metastasis effect and their mechanism as well. BP isolated from Radix *Angelica sinensis* is reported to suppress the metastasis of bladder cancer. Human bladder cancer cells (5637, BFTC, T24, and TCCSUP) were treated with 60 μg/mL BP for 24 h. In turn, BP up-regulated E-cadherin and down-regulated N-cadherin, so the result implicates that BP inhibit the migration in bladder cancer cells via the modulation of E-cadherin and N-cadherin.

Coccia et al. reported the anti-metastatic effect of extra virgin olive oil (EVOO) phenols [72]. Normal urothelial fibroblasts were tested in the same condition with T24 cells to investigate whether the EVOO extract exerts an anti-metastatic effect on tumor cells without affecting normal urothelial fibroblasts. The EVOO extract suppressed migration and invasion of the T24 cells, not the fibroblasts by attenuating MMP-2, not the MMP-9 expression levels and increasing the TIMP-1, TIMP-2 expression levels.

Shin et al. revealed that garlic inhibits cell migration and invasion [73]. Garlic (*Allium satibum* L.) is a perennial bulb plant that contains organic sulfur compounds such as alliin, allicin, ajoene, and diallyl polysulfides. The garlic extract (GE) impeded the migration and invasion of the EJ cells via inhibition of the MMP-9 expression followed by the decreased binding activities of the AP-1, Sp-1, and NF-κB motifs. HSPA6, the most up-regulated gene in the GE-treated EJ cells, was verified to suppress transcription factor-associated MMP-9 regulation.

Cheng et al. reported that sinulariolide inhibits cell migration and invasion [74]. Sinulariolide is a natural product extracted from the cultured-type soft coral *Sinularia flexibilis* and possesses bioactivity against the movement of bladder cancer cell. TSGH-8301 cells exposed to sinulariolide showed decreased protein expression levels of MMP-2/-9, urokinase, and increased levels of TIMP-1/-2. Moreover, the expressions of cell migration- and invasion-related proteins (GRB2, Ras, RhoA, MKK3, and MKK7) were repressed.

**Table 4 pharmaceutics-15-01117-t004:** Natural products inhibiting metastasis.

Classification	Compound/Extract	Source	Cell Line/Animal Model	Dose; Duration	Efficacy	Mechanism	Ref.
Single compound	Flaccidoxide-13-acetate	*Cladiella kashmani*	RT4, T24	2.5, 5, 10 μM; 24 h	Inhibition of cell migration and invasion	↑TIMP-1, TIMP-2 ↓MMP-2, MMP-9, uPAR, FAK, PI3K, p-PI3K, AKT, p-AKT, mTOR, p-mTOR, Rho A, Ras, MKK7, MEKK3	[75]
Single compound	N-butylidenephthalide	*Radix Angelica sinensis*	5637, BFTC, T24, TCCSUP	60 μg/mL; 24 h	Inhibition of metastasis	↑E-cadherin ↓N-cadherin	[33]
Single extract	Extra virgin olive oil extract (EVOO-E)	*Olea europaea* L. var. Itrana	T24	2.5, 10 μg/mL; 24 h	Inhibition of cell migration and invasion	↑TIMP-1, TIMP-2 ↓MMP-2	[72]
Single extract	Garlic extract (GE)	*Allium satibum* L.	EJ	400, 600, 800 μg/mL; 24 h	Inhibition of cell migration and invasion	↓MMP-9, AP-1, Sp-1, NF-κB	[73]
Single extract	Sinulariolide	*Sinularia flexibilis*	TSGH-8301	800 μg/mL; 12, 24 h	Inhibition of cell migration and invasion	↑HSPA6	[74]

Footer: ↑, up-regulation; TIMP, metalloproteinase; ↓, down-regulation; MMP, matrix metalloproteinase; uPAR, urokinase-type plasminogen activator receptor; FAK, focal adhesion kinase; PI3K, phosphatidylinositide-3 kinases; AKT (PKB), protein kinase B; mTOR, mammalian target of rapamycin; Rho A, Ras homolog gene family, member A; Ras, regulators of signal transduction; MKK, mitogen-activated protein kinase kinase; MEKK (MAPKKK or MAP3K), Mitogen-Activated *Protein* Kinase Kinase Kinase; BFTC, Black Foot disease Transitional *Carcinoma*; TCCSUP, isolated from an anaplastic transitional *cell* carcinoma (TCC) in the neck of the urinary bladder; AP, activator protein; Sp, specificity protein; NF-κB, nuclear factor kappa-light-chain-enhancer of activated B cells; TSGH, human gastric carcinoma *cell line*; HSPA6, Heat shock protein A6.

Neoh et al. presented various background knowledge to support the main theme that metastasis is responsible for most of the cancer deaths and that the results of the suggested studies show that diterpenoids have cytotoxicity against cancer cells [75]. Moreover, it was meaningful that they utilized immunostaining to measure the expressions of proteins so as to further understand the effects of flaccidoxide-13-acetate.

Chiu et al. used nine methods in total including cell culture, TUNEL assay, annexin V-FITC staining, patients, and study design and specifically suggested the reasons to the usage of each method [33]. Moreover, it was unique to further suggest the synergistic cytotoxic effect of BP in combination with cisplatin. The chemical structures of the compounds were presented (Figure 3) [43]. The inhibiting metastasis of the natural products were elucidated in Figure 4.

## 6. Natural Products and Angiogenesis

Dis-regulated angiogenesis is responsible for solid tumor growth and metastasis. The angiogenesis switch increases the production of vascular endothelial growth factor (VEGF) by the activation of hypoxia-inducible transcription factor [77]. Three studies indicated natural products that inhibit the angiogenesis of bladder cancer cell through hindering the vascular endothelial growth factor (VEGF) pathway (Table 5).

The acetone extract of *Angelica sinensis* (AE-AS) showed an antiangiogenic activity [78]. AE-AS inhibited the tube formation of human umbilical vascular endothelial cells (HUVECs). AE-AS also diminished the angiogenesis in chicken chorioallantoic membrane (CAM), which is stimulated by the hypoxia and T24 cell. AE-AS suppressed vasculature formation in Matrigel plug. During the antiangiogenic process, WSB-1, pVHL, HIF-1α, VEGF, and the VEGFR2 cascade was down-regulated.

The mechanism underlying the anti-angiogenesis effects of green tea polyphenol (GTP) in bladder cancer cells is reported [79]. N-butyl-N-(4-hydroxybutyl) nitrosamine (BBN) solution was treated on C3H/He mice for 14–24 weeks, and in the other group, 0.5% GTP solution was additionally treated for the same period. In the BBN group, cytoplasmic HuR expression was higher compared with the control group and showed activated invasion of tumor cells. However, cytoplasmic HuR was not expressed in the BBN + GTP group. Furthermore, according to their analysis, GTP was associated with COX-2 and HO-1 expression, which is related with cell proliferation. These results show that GTP has anti-proliferative and anti-angiogenic effect on tumor cells.

Chen et al. conducted experiments systematically through several experiments [78]. They indirectly demonstrated the effect of AE-AS on inhibiting angiogenesis but showed how indirect materials actually affect angiogenesis inhibition. In addition, they performed several assays such as quantitative real-time PCR assay, western blotting assay, co-immunoprecipitation assay, etc.

It is excellent in that Matsuo et al. investigated a present study because there is no mechanistic basis for the previously known effect of GTP on inhibiting angiogenesis [79]. The chemical structures of the compounds were presented (Figure 5) [43]. The inhibiting angiogenesis of the natural products were elucidated in Figure 6.

**Table 5 pharmaceutics-15-01117-t005:** Natural Products inhibiting angiogenesis.

Classification	Compound/Extract	Source	Experimental Model	Dose; Duration	Efficacy	Mechanism	Ref.
Single extract	Acetone extract of RAS (AE-AS)	*Angelica sinensis*	Hypoxia-treated T24	10, 30, 40 μg/mL; 8 h	Inhibition of angiogenesis	↑pVHL ↓HIF-1α, VEGF, VEGFR2, PI3K, AKT, mTOR, WSB-1	[78]
			T24 bearing BALB/c mice	100, 250, 500 mg/kg/day; 30 days		↓HIF-1α, VEGF, CD31, p-VEGFR2	
Single extract	Green tea polyphenol (GTP)	*Camellia sinensis* O. Kuntze	C3H/He mice	0.5% GTP in tap water; 14, 24 weeks	Inhibition of angiogenesis	↓C-HuR, HO-1, VEGF-A	[79]
β-carboline alkaloids	Harmine	*Pergamum harmala* seeds	RT4 bearing BALB/c mice	10 mg/kg/day; 30 days	Inhibition of angiogenesis	↓p-VEGFR2	[80]

Footer: RAS, root of Angelica sinensis; BALB, Bagg and Albino; ↑, up-regulation; pVHL, von Hippel-Lindau tumor suppressor; ↓, down-regulation; HIF-1α, hypoxia-inducible factor-1α; VEGF, vascular endothelial growth factor; VEGFR2, VEGF-receptor-2; PI3K, phosphatidylinositide-3 kinases; AKT (PKB), protein kinase B; mTOR, mammalian target of rapamycin; WSB-1, WD repeat and SOCS box-containing protein-1; CD31, cluster of differentiation 31; C-HuR, Cytoplasmic human antigen R; HO, hemeoxygenase.

## 7. Natural Products and Drug Resistance

Drug resistance, or chemoresistance, is a significant obstacle to the successful treatment of bladder cancer. More active research on strategies that lead to reversion of drug resistance is essential to improve the control of bladder cancer. There is a study that reported a natural compound that has a reversal effect on drug resistance and its underlying mechanisms (Table 6).

Resveratrol (RES) shows the reversal effect in pumc-91/ADM cells on multidrug resistance [81]. Unlike RES, (−)-epigallocatechin gallate and ginsenoside Rh2, only RES at doses of 10, 50 and 100 µM had a reversal effect on the ADM resistance of pumc-91/ADM cells. RES-treated pumc-91/ADM cells showed cell cycle arrest at S phase, decreased MRP1, LRP, GST, BCL-2 expression levels and increased Topo-II levels. The chemical structures of the compounds were presented (Figure 7) [43]. The sensitizing drug resistance of the natural products were elucidated in Figure 8.

## 8. Clinical Trials

The seven papers refer to experiments to see the efficacy of compounds or extracts such as erlotinib, green tea extract, broccoli sprout extract, cranberry fruit juice extract (Table 7).

AbnobaVISCUM 900 is compared to mitomycin C in efficacy [82]. A total of 546 participants would provide written informed consent before any study related procedures will be performed. But nothing is found yet, because this project’s recruitment status is still ongoing.

AbnobaVISCUM Fraxini is investigated for intravesical instillation in a total of 37 participants with superficial bladder cancer and identified the local and systemic tolerability, the influence on tumor remission, and the influence on the one-year recurrence rate [83]. This project’s recruitment status is completed. It was found that abnobaVISCUM Fraxini shows a direct antitumoral effect.

Broccoli sprout extract shows side effects treating patients with transitional cell bladder cancer undergoing surgery [84]. A total of 7 participants received oral broccoli sprout extract once daily on days 1–14 in the absence of disease progression or unacceptable toxicity. This project’s recruitment status is terminated. It was found that broccoli sprout extract prevents or slows the growth of certain cancers.

Erlotinib together with green tea extract is studied to prevent cancer recurrence in former smokers who have undergone surgery for bladder cancer [85]. A total of 17 participants will be accrued for this study within 3 years. This project’s recruitment status is completed. It was found that erlotinib and green tea extract kill any remaining tumor cells and prevent the recurrence of bladder cancer.

Fluorescent dye predicts the cancer’s invasiveness [86]. This trial studies an ultrasound test and a biomarker test. The purpose of this study is to check how well the two tests predict the aggressiveness of bladder cancer. A total of 4 participants with bladder cancer participated in the test.

Green tea catechin extract is studied to compare to a placebo when given before surgery in treating a total of 31 participants with nonmetastatic bladder cancer [87]. This project’s recruitment status is completed. It was found that green tea catechin extract slows the growth of certain cancers.

Urell together with cranberry fruit juice extract and proanthocyanidins prevents uropathogenic *E. coli* bacteria from adhering to uroepithelial cells [88]. This single group assignment is due to the intervention itself and the duration of postoperative ureteral and bladder, and it leads to an increase in surveillance and antibiotic treatment, given the risks of declared infection in this context, which is terminated. The chemical structures of the compounds were presented (Figure 9) [43].

Through experiments with extracts, it was found that broccoli sprout extract inhibits the growth of certain cancers [84].

However, some trials have not posted specific mechanisms [82,83,85,86,88]. The clinical trial related to fluorescent dye terminated early when the PI left the institution, so the accuracy of the results is low [86]. It was found that Urell inhibits uropathogenic *E. coli* bacteria from adhering to uroepithelial cells [88]. However, in the perioperative setting of radical cystectomy-bladder replacement, no improvement of UTI was noticed in 10 patients who took Urell following the protocol. However, this clinical trial terminated early despite its low accuracy. It was also a problem that the number of experimenters participating in clinical trials was small.

**Table 7 pharmaceutics-15-01117-t007:** Clinical Trial.

Compound/Extract	Source	Phase	Participants	Status	Registration Number	Results	Ref.
abnobaVISCUM 900, Mitomycin C	*Viscum album* var. coloratum, Mitomycin C	III	546	Recruiting	NCT02106572		[82]
abnobaVISCUM Fraxini	*Viscum album* var. coloratum	I, II	37	Completed	NCT02007005	Indication of a direct anti-tumoral effect	[83]
Broccoli sprout extract	*Brassica oleracea* var. italica	NA	7	Terminated	NCT01108003	Inhibition of the growth of certain cancers	[84]
Erlotinib, Green tea extract	Erlotinib hydrochloride, *Camellia sinensis*	II	17	Completed	NCT00088946	Inhibition of any remaining tumor cells and inhibition of the recurrence of bladder cancer	[85]
Fluorescent dye	calcium dye	NA	4	Terminated	NCT02494635	Prediction of the cancer invasiveness	[86]
Green tea catechin extract	*Camellia sinensis*	II	31	Completed	NCT00666562	Inhibition of the growth of certain cancers	[87]
Urell, Cranberry fruit juice extract, proanthocyanidins	Proanthocyanidins, Vaccinium microcarpum	NA	10	Terminated	NCT03986398	Inhibition of uropathogenic E.col bacteria from adhering to uroepithelial cells	[88]

## 9. Compounds Whose Names Are Unknown

The exact names of some compounds were unknown. Molecular weight is added for the distinction as follows.
pentacyclic oxindole alkaloids (C_21_H_24_N_2_O_4_)OAPF (oxindole alkaloid purified fraction)OAPF 15′(oxindole alkaloid purified fraction after 15 min heating under reflux)OAPF 45′(oxindole alkaloid purified fraction after 45 min heating under reflux)OAPF 5′(oxindole alkaloid purified fraction after 5 min heating under reflux)Triterpenoid saponin tub (C_65_H_102_O_29_)Triterpenoid saponin **1**Triterpenoid saponin **2**Triterpenoid saponin **4**Triterpenoid saponin **5**Saponin (C_58_H_94_O_27_)Saponin **1**Saponin **2**Saponin **3**Saponin **4**Saponin **12**Saponin **13**

## 10. Discussion

Bladder cancer (BC) is a disease that has a variable natural history [1]. Bladder cancer can be defined as a tumor developed in the bladder lining. Its characteristic is known as high incidence, morbidity, and mortality [4]. Chemotherapy and immunotherapy are commonly used to treat bladder cancer. However, chemotherapy can cause burning and irritation in the bladder. In addition, BCG immunotherapy, which is the main type of intravesical immunotherapy for bladder cancer, can also cause burning in the bladder and flu-like symptoms [6,7]. Thus, it is essential to develop drugs originating from natural products. This paper focused on the natural products that suppress or treat bladder cancer.

In this study, 87 papers that dealt with natural products preventing or treating bladder cancer were reviewed. The studies were classified into the following mechanism: 71 papers on cell death, 5 papers on anti-metastasis, 3 papers on anti-angiogenesis, 1 paper on anti-resistance, and 7 papers on clinical trials. More in vivo studies and clinical trials should be conducted. Materials that are well-studied including in vivo studies, whose mechanism is revealed, and whose effective concentration is low, should be studied in clinical trials.

### 10.1. Natural Products Inducing Apoptosis on Bladder Cancer Cells

Cell death includes apoptosis, necrosis, oncosis, pyroptosis, and autophagy. Among the several types of cell death, apoptosis was the most studied type. Apoptosis is the process by which a cell divides and proceeds to the controlled death of the cell [14]. Apoptosis is dependent on the activation of initiator caspases and executioner caspases, resulting DNA fragmentation [14]. Ethanol extract of pomegranate fruit induced apoptosis on T24 cells by up-regulating pro-caspase-3, -8,-9,-12 Bax/Bcl-2 ratio, CHOP, and Bip [61]. Regulation of caspase-3, caspase-8, and caspase-9 lead to the mitochondrial damage and death receptor signaling. Regulation of pro-caspase-12 lead to the endoplasmic reticulum (ER) stress. Furthermore, CHOP and Bip, which are the ER stress markers, increased after the treatment. N-butylidenephthalide (BP) from *Radix Angelica sinensis* induced apoptosis on BFTC by activating caspase-3, -9, E-cadherin, and deactivating N-cadherin [33]. Activation of caspase-3 and caspase-9 induced mitochondria-mediated apoptosis. Treatment of ethanol extracts of peel (EECU) on T24 cells up-regulated ROS, Caspase-8, -9, -3 and down-regulated P13K, Akt, LY294002 [49]. Activation of caspase-8 and caspase-9 activates effector caspases such as caspase-3, which lead to apoptosis by causing degradation of substrate proteins such as DNA repair enzyme. Treatment of Guizhi Fuling Wan extract on BFTC 905 and TSGH 8301 induced apoptosis by up-regulating CHK2, p21, and down-regulating CDK2-cyclin E, A, and A complexes [53]. Phosphorylated CHK2 promotes the expression of p21, resulting in the interference of the cell cycle progression and the induction of apoptosis.

### 10.2. Natural Products Inducing Anti-Angiogenesis Effect on Bladder Cancer Cells

Angiogenesis is the process of development of new blood vessels from other blood vessels [89]. It plays an important role in tumor growth, progression, and metastasis. Anti-angiogenesis strategies include anti-angiogenetic drug, chemotherapy, or immunotherapy [90]. Treatment of AE-AS on Hypoxia-treated T24 cells and T24 bearing BALB/c mice inhibited the angiogenesis both in vivo and vitro [78]. Significant increase and activity of HIF-1α occurred in the tumor hypoxic microenvironment is a crucial force to trigger angiogenesis. AE-AS suppressed WSB-1-dependent pVHL degradation and inhibited ROS production, which led to the down-regulation of HIF-1α. Treatment of green tea polyphenol (GTP) on C3H/He mice inhibited angiogenesis, by down-regulating C-HuR, HO-1, and VEGF-A [79]. GTP intake suppressed cytoplasmic HuR expression, and this led to the inactivation of HO-1 and VEGF-A, which are the angiogenic factors. In addition, GTP intake directly inactivated HO-1.

### 10.3. Natural Products Inducing Anti-Metastasis Effect on Bladder Cancer Cells

Cancer metastasis is a complex disease, arising from a growing tumor from which cells escape to other parts of the body [91]. Treatment of flaccidoxide-13-acetate on RT4, T24 cells suppressed cell migration and invasion by down-regulating FAK, PI3K, AKT, mTOR, MMP-2, and MMP-9 [75]. The treatment reduced the activity of FAK/PI3K/AKT/mTOR signaling proteins. Then, the inactivated FAK/PI3K/AKT/mTOR signaling proteins down-regulated the metastasis-related protein MMP-2 and MMP-9, resulting in the inhibition of cell migration and invasion. Sinulariolide inhibited the migration and invasion of TSGH-8301 cell by down-regulating MMP-2/-9 and urokinase, and up-regulating TIMP-1/-2 protein [74]. The mechanism of the signaling pathway involved in the inhibitory effect of sinulariolide was a reduction in phosphorylated FAK, PI3K, AKT, and mTOR proteins.

### 10.4. Natural Products Repressing Drug Resistance on Bladder Cancer Cells

Drug resistance is the reduction in the efficacy and potency of a drug to make therapeutic advantages. It is a big obstacle to the treatment and the survival of patients [92]. RES reversed ADM resistance in pumc-91/ADM cells by increasing Topo-II levels and decreasing MRP1, LRP, GST, BCL-2 levels [81].

### 10.5. Promising Substance for the Clinical Trial

Sulforaphane, a compound extracted from broccoli sprout, is a promising candidate for the clinical trial. It is well-studied including in vivo studies, and its anti-bladder cancer mechanism is studied as well. Treatment of 20 μM sulforaphane on UMUC3 cells for 48 h down-regulated survivin 35%, HER2/neu 90%, and EGFR 54% in. Tumor growth rates in UMUC3 bearing female athymic mice showed a 42% inhibition by sulforaphane treatment at a dose of 295 μmol/kg/day for 2 weeks with decreased Ki67-positive cells [22]. Athymic mice were injected subcutaneously with a UM-UC-3 cell, and sulforaphane was treated at doses of 12 mg/kg body weight for 5 weeks [38]. The average tumor volume decreased by the inhibitory rate of 63%. In the sulforaphane treated mice, expression of caspase 3 and cytochrome c were induced, and the expression of survivin was reduced. 

### 10.6. Cautions on Using Natural Products

Nevertheless, certain natural products that interact with chemotherapy during chemotherapy should not be taken [93]. For instance, *Euphorbia pekinensi* (*EP*)*s*’s diterphenoid is known to be anti-tumor as it has a cytotoxic effect, but according to “18 ban 19 eoi” [94] which are taboos against taking traditional medicine, *Euphorbia pekinensis* cannot be combined with *Glycyrrhizae radix et rhizome* (*GR*).

### 10.7. Limitations

Wu et al. reviewed the anti-bladder cancer effect of natural products of plant origin [95]. Wigner et al. reviewed the role of natural compounds in bladder cancer treatment [96]. However, they did not include the effective dose and duration of the compound, and they only reviewed single compounds that had anti-bladder cancer effects. In contrast, this study contains the effective dose and duration of all the natural products included. On top of that, the present study includes not only single compounds, but also extracts of natural products.

There were some limitations on our study and the papers reviewed. Our study only covered papers that are published in the last 8 years (2016–2023) and only reviewed papers that are written in English. The reviewed papers mostly had an experiment in vitro and lacked human study. More human studies regarding the anti-bladder cancer effect of natural products should be conducted. There were some papers that lacked duration of the treatment of natural products [19,27].

## 11. Conclusions

The present study reviewed 87 papers that showed the anti-bladder cancer effect of natural products. Only papers published from 2016 to 2023 and written in English were reviewed. This study classified the papers into the following mechanisms: cell death, anti-metastasis, anti-angiogenesis, anti-resistance, and clinical trials. Natural products had a significant anti-cancer effect on various mechanisms. Moreover, molecules such as caspase-3, caspase-9, MMP-2, MMP-9, HIF-1α, and VEGF-A were frequently regulated. However, the number of papers regarding anti-resistance and clinical trial were too few, so more study should be conducted. Ultimately, this database will be useful for the future in vivo studies of the anti-bladder cancer effect of natural products, in the process of selecting materials used for the experiment.

## Figures and Tables

**Figure 1 pharmaceutics-15-01117-f001:**
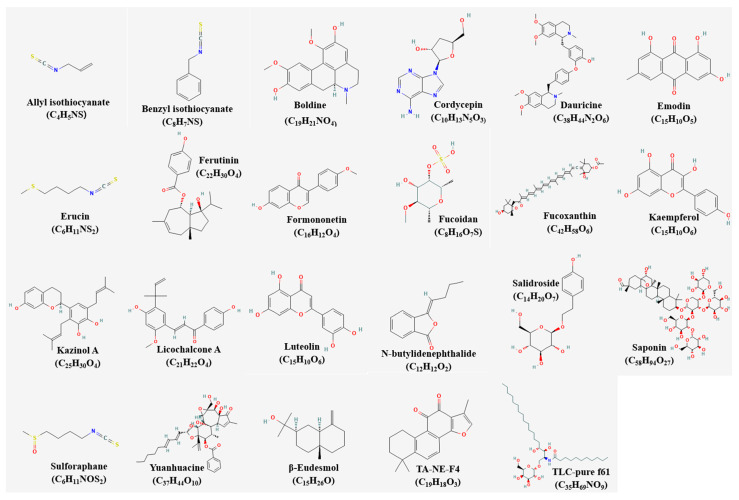
Chemical structures of compounds inducing apoptosis.

**Figure 2 pharmaceutics-15-01117-f002:**
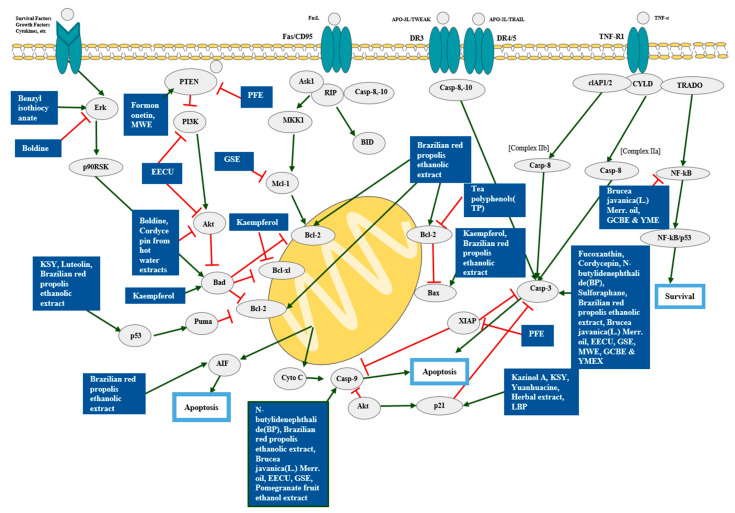
Schematic diagram of apoptosis. After factors such as FasL, APO-3L, APO-2L, and TNF-α attaching to the receptors located in cell membrane, apoptosis begins. Bad and Mcl-1 protein regulate Bcl-2 family, which control release of proteins from the space between the mitochondrial inner and outer membrane. Then, Bcl-2 family regulates proteins such as Cyto C, and Cyto C activates caspase-9. Finally, Caspase-9 induces apoptosis [68,69]. On the other hand, apoptosis also occurs by caspase-3, which is activated by caspase-8 and caspase-10. Various natural products induced apoptosis by regulating key molecules of apoptosis. Abbreviation: AIF, Apoptosis-inducing factor; Akt, Protein kinase B, Ask1, Apoptosis signal-regulating kinase 1; APO-3L/TWEAK, APO3 ligand or TNF-related weak inducer of apoptosis; APO-2L/TRAIL, Apo2 ligand or tumor necrosis factor-related apoptosis-inducing ligand; Bad, BCL2 associated agonist of cell death; Bax, Bcl-2 Associated X-protein; Bcl-2, B-cell lymphoma 2; BID, BH3 Interacting Domain Death Agonist; Casp-8, Caspase-8; Casp-10, Caspase-10; Casp-9, Caspase-9; Casp-3, Caspase-3; CD95, cluster of differentiation 95; clAP1/2, cellular inhibitor of apoptosis 1 and 2; CYLD, *Cylindromatosis*; Cyto C, *Cytochrome c*; DR3, Death Receptor 3; DR4/5 Death receptor 4, 5; EECU, Ethanol extracts of peel; Erk, Extracellular signal-regulated kinase; FasL, Fas ligand; KSY, Kuan-Sin-Yin; GCBE, green coffee bean extract; LBP, Lycium barbarum polysaccharides; Mcl-1, Myeloid cell leukemia 1; MKK1, Mitogen-activated protein kinase kinase 1; MWE, Mulberry Water Extract; NF-kB, Nuclear factor kappa B; p90RSK, p90 ribosomal S6 kinase; p53; PTEN, phosphatase and tensin homolog; PFE, Pomegranate fruit ethanol extract; PI3K, Phosphoinositide 3-kinase; Puma, p53 up-regulated modulator of apoptosis; RIP, Receptor-interacting protein; TNF- α, *tumor necrosis factor*-*α*; TNF-R1, Tumor necrosis factor receptor 1; TRADO; GSE, Grape seed extract; XIAP, X-linked inhibitor of apoptosis protein; YME, yerba mate extract; YMEX, yerba mate extract. Reprinted/adapted with permission from Ref. [70]. 2013, ApexBio Technology.

**Figure 3 pharmaceutics-15-01117-f003:**
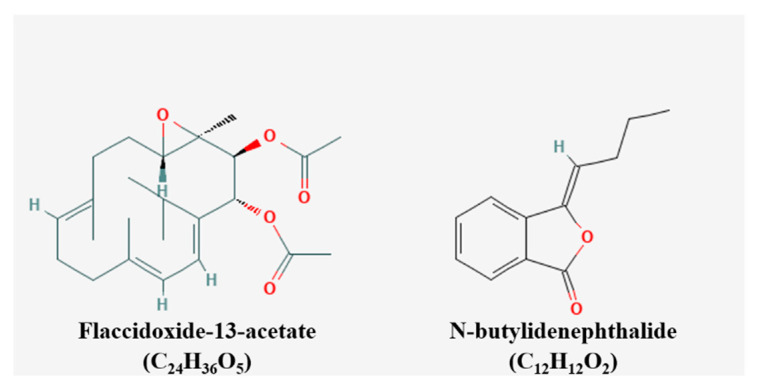
Chemical structures of compounds inhibiting metastasis.

**Figure 4 pharmaceutics-15-01117-f004:**
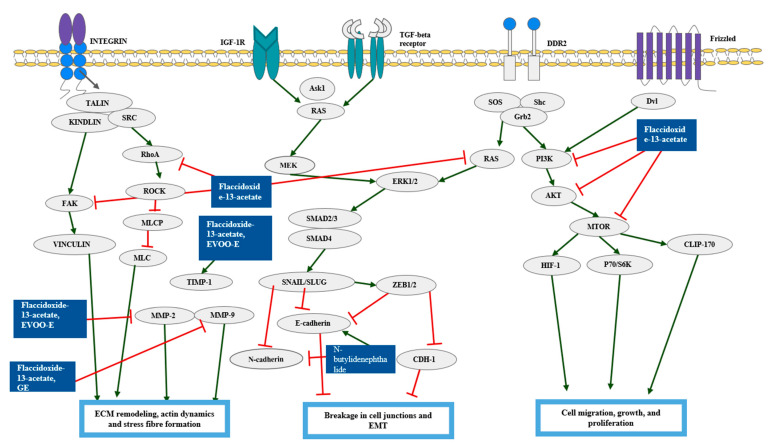
Schematic diagram of metastasis. ECM is a highly organized structure where cells can follow the matrix fibers and migrate. By remodeling it, cells can move to nearby tissues. ECM remodeling is regulated by factors such as MMP-2 and MMP-9. Actin dynamics is essential for the Epithelial-to-Mesenchymal Transition (EMT), which gives cells the migratory potential. ZEB1/2 is a key molecule regulating EMT. HIF-1 regulates the genes and cause metastasis. Natural products were capable of suppressing the molecules related to the metastasis. Abbreviation: Ask1, Apoptosis signal-regulating kinase 1; AKT, Protein kinase B; CDH-1, Cadherin-1; CLIP-170, Cytoplasmic Linker Protein; Dvl, disheveled; DDR2, Discoidin Domain Receptor 2; ERK1/2, Extracellular signal-regulated kinase 1, 2; E-cadherin, epithelial cadherin; N-cadherin, neural cadherin; EVOO-E, Extra virgin olive oil extract; FAK, Focal Adhesion Kinase; GRB2, Growth factor receptor-bound protein 2; GE, Garlic extract; HIF-1, Hypoxia-Inducible Factor-1; IGF-1R, Insulin-like Growth Factor-1R; MLCP, Myosin Light-chain phosphatase; MLC, Mixed Lymphocyte Culture; MMP-2, Matrix metalloproteinase-2; MMP-9, Matrix metalloproteinase-9; MTOR, Mammalian Target of Rapamyci; PI3K, Phosphoinositide 3-kinase; P70/S6K, Ribosomal protein S6 kinase beta-1; RhoA, Ras homolog family member A; ROCK, Rho-associated Protein Kinase; RAS, Renin-angiotensin system; SRC, Steroid Receptor Coactivator; SMAD2/3, Suppressor of Mothers against Decapentaplegic 2, 3; SMAD4, Suppressor of Mothers against Decapentaplegic 4; SOS, the Son of Sevenless; Shc, SHC-adaptor protein; Talin and Kindlin, two families of FERM-domain proteins that bind the cytoplasmic tail of integrins; TIMP-1, Tissue inhibitor of matrix metalloproteinase-1; TGF, Transforming Growth Factor; Vinculin, a cytoskeletal protein associated with cell-cell and cell-matrix junctions; ZEB1/2, zinc finger E-box binding homeobox-1,2, reprinted/adapted with permission from Ref. [76]. 2021, Bikashita Kalita, Mohane Selvaraj Coumar.

**Figure 5 pharmaceutics-15-01117-f005:**
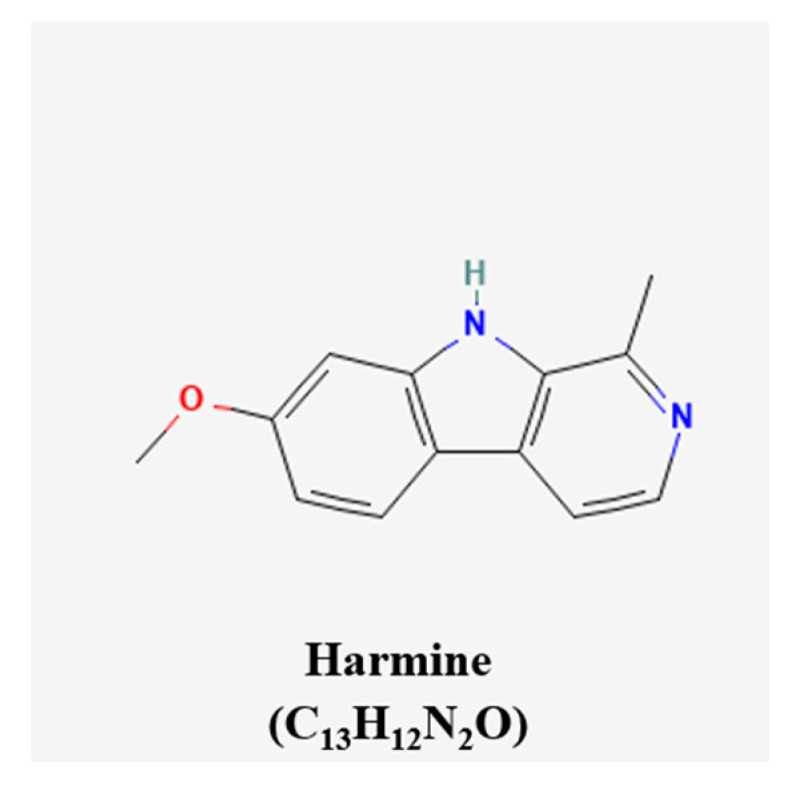
Chemical structure of compound inhibiting angiogenesis.

**Figure 6 pharmaceutics-15-01117-f006:**
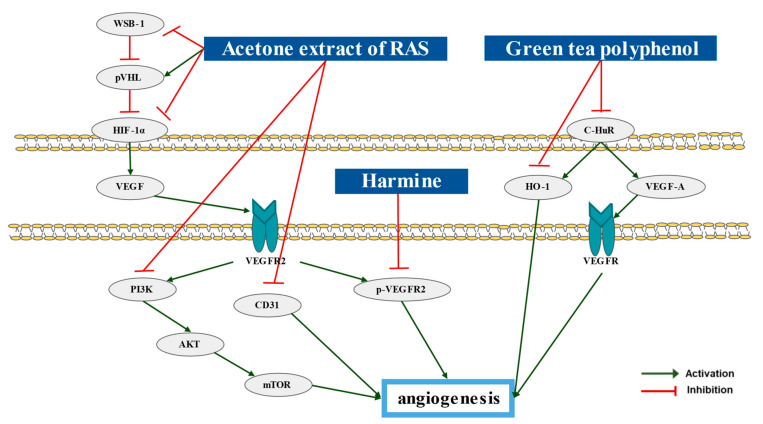
Natural products and angiogenesis. HIF-1a promotes VEGF expression to activate VEGFR2. Due to activated VEGFR2, PI3K, AKT, and mTOR expression levels are sequentially increased, resulting in angogenesis. Activated VEGFR2 also promotes p-VEGFR2 expression and causes angogenesis. AE-AS inhibits all substances in this process. It also increases pVHL expression inhibiting HIF-1a, the beginning of the process, and re-inhibits WSB-1 inhibiting pVHL, and harmine inhibits p-VEGFR2 resulting in angiogenesis. Moreover, C-HuR causes angiogenesis. GTP intake suppresses the expression of HO-1 directly and that of HO-1 and VEGF-A indirectly via regulation of C-HuR expression. Abbreviation: AKT (PKB), protein kinase B; CD31, cluster of differentiation 31; C-HuR, Cytoplasmic human antigen R; HIF-1α, hypoxia-inducible factor-1α; HO-1, hemeoxygenase-1; mTOR, mammalian target of rapamycin; PI3K, phosphatidylinositide-3 kinases; p-VEGFR2, protein-VEGFR2; pVHL, p-von Hippel-Lindau tumor suppressor; RAS, root of Angelica sinensis; VEGF, vascular endothelial growth factor; VEGFR2, VEGF-receptor-2; WSB-1, WD repeat and SOCS box-containing protein-1.

**Figure 7 pharmaceutics-15-01117-f007:**
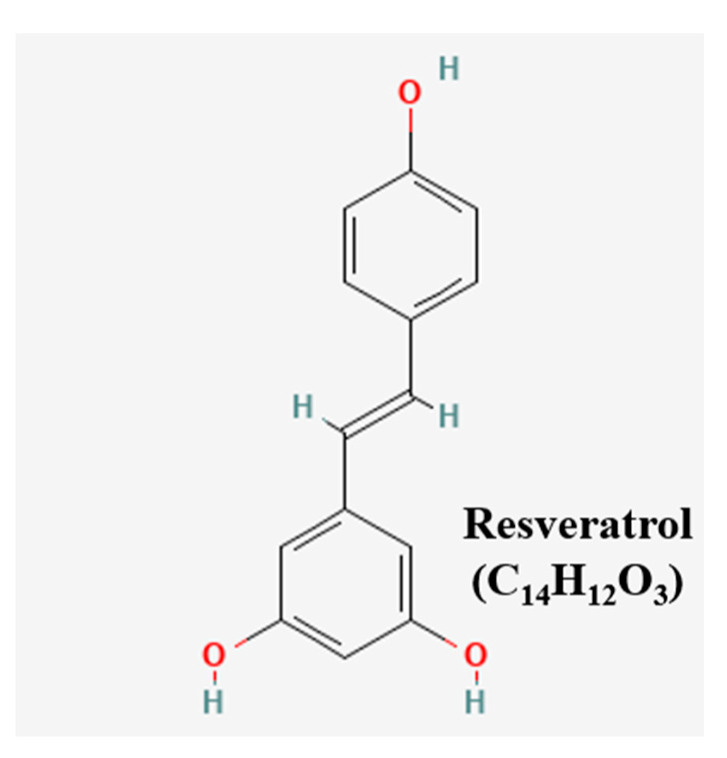
Chemical structure of compound reversing drug resistance.

**Figure 8 pharmaceutics-15-01117-f008:**
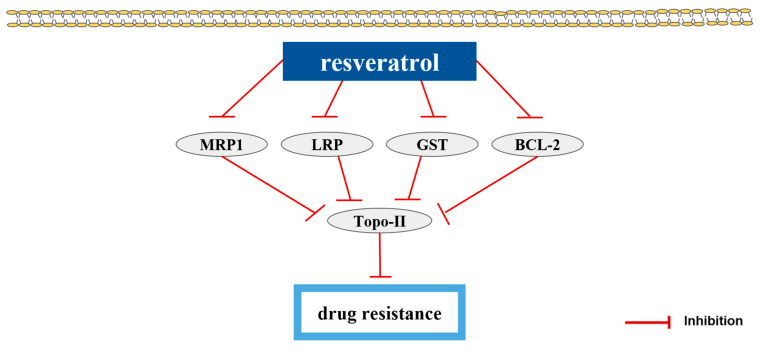
Natural products and drug resistance. Topo-II suppresses drug resistance. RES decreases MRP1, LRP, GST, BCL-2 expression levels. By the law of total mass conservation, Topo-II levels are increased. As a result, RES inactivates drug resistance. Abbreviation: BCL-2, B-cell leukemia/lymphoma 2; LRP, lung resistance protein; MRP1, multi-drug resistance 1; Topo-II, topoisomerase II; GST, glutathione S-transferase.

**Figure 9 pharmaceutics-15-01117-f009:**
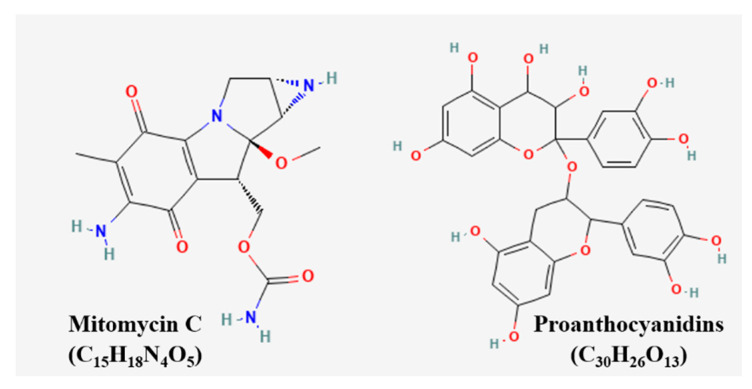
Chemical structures of compounds used in the clinical trials.

**Table 1 pharmaceutics-15-01117-t001:** Single Natural Compounds inducing Apoptosis.

Compound/Extract	Source	Experimental Model	Dose; Duration	Efficacy	Mechanism	Ref.
5,7-Dimethoxy-8-[(Z)-3′-methylbutan-1′,3′-dienyl]coumarin	*Murraya tetramera*	EJ	IC_50_ 30.59 µg/mL; 48 h	Inhibition of proliferation		[15]
Allyl isothiocyanate	*Lepidium latifolium* L.	UM-UC-3	IC_50_ 23.27 μg/mL; 24 h	Induction of cytotoxicity		[16]
Benzyl isothiocyanate		5637	20 µM; 24 h	Induction of miR-99a expression	↑p-JNK, p-ERK, p-c-Jun, c-Jun/AP-1, ERK, miR-99a	[17]
		T24			↑p-Akt, p-JNK, p-ERK, p-c-Jun, c-Jun/AP-1, ERK, miR-99a	
Boldine	*Peumus boldus*	T24	200, 300, 400, 500 μM; 24 h	Induction of apoptosis	↑GSK-3β ↓ERK, AKT	[18]
Cordycepin from hot water extracts (cordycepin-CMHW)	*Cordyceps militaris*	T24	0, 10, 30, 90 μg/mL; duration is not given	Induction of apoptosis	↑Caspase-3, A3AR ↓ Akt, Lef/Tcf, C-myc, Cyclin D1	[19]
Dauricine	*Menispermum dauricum*	EJ	8, 16, 32, 64 μg/mL; 24, 48, 72 h	Inhibition of proliferation		[20]
Emodin	*Frangula alnus*	MBT2, T24, TSGH8301, J82	40, 80 µM; 24 h	Inhibition of inflammation Inhibition of proliferation	↑H3K27me3 ↓pH3Ser10	[21]
		T24	40 µM; 24 h		↓FABP4, HBP17	
		T24	40 µM; 24 h		↓RGS4	
Erucin	*Brassica oleracea var. italica*	UMUC3	20 μM; 48 h 295 μmol/kg/day; 2 weeks	Induction of apoptosis Suppression of cancer cell growth	↓survivin, EGFR, HER2/neu ↓Ki67	[22]
		UMUC3 bearing female athymic mice				
Ferutinin	*Ferula ovina*	TCC	IC_50_ 24 μg/mL; 72 h	Inhibition of proliferation		[23]
			IC_50_ 33 μg/mL; 24 h			
Formononetin from *Astragalus membranaceus*	*Astragalus membranaceus*	miR-21, PTE N	50, 100, 200 μM/mL; 48 h	Inhibition of proliferation and invasiveness	↑PTEN ↓miR-21, p-AKT	[24]
Fucoidan from *Fucus vesiculosus*	*Fucus vesiculosus*	5637	100 μg/mL; 6, 12, 24, 48 h; 10, 25, 50, 100 μg/mL; 24 h	Induction of apoptosis	↑ROS, Bax/Bcl-2 ratio	[25]
Fucoxanthin	*Undaria pinnatifida*	T24	20, 40 μM; 48 h	Induction of apoptosis	↑Mutant-type p53, caspase-3 ↓Mortalin-p53	[26]
	*Cassia alata* L.	T24	25, 50, 100 µg/ mL IC_50_ 17.13 µg/mL 50 µg/mL; 12–24 h	Induction of cytotoxicity		[27]
Kaempferol		EJ	20, 40, 80 μM; 48 h	Induction of apoptosis	↑p-p53, Bax, Bad ↓Bid, Mcl-1, Bcl-xL, p-AKT	[28]
Kazinol A from *Broussonetia papyrifera*	*Broussonetia papyrifera*	SW620, MCF-7, T98G, T24, T24R2, HEK293	20 μM; 24 h	Induction of cytotoxicity	↑p21 ↓Cyclin D1	[29]
Kuan-Sin-Yin (KSY)	*Codonopsis pilosula* (*Franch.*) Nannf., *Poria cocos* (*Schw.*) Wolf, *Atractylodes macrocephala* Koidez., *Glycyrrhiza uralensis* Fisch., *Ligustrum lucidum* Ait., *Astragalus membranaceus* (*Fisch.*) Bunge., *Pogostemon cablin* (*Blanco*) *Bench*.	MBT-2	1500, 2000 μg/mL; 72 h	Inhibition of proliferation		[30]
		MBT-2 bearing C3H/HeN	1000 mg/kg/day; 15 days	Suppression of cancer cell growth		
		MBT-2 bearing BALB/cAnN-Foxn1nu/Cr1Nar1	1000 mg/kg/day; 15 days	Suppression of cancer cell growth	↑p53, p21 ↓Ki-67	
Licochalcone A (LCA)	*Glycyrrhiza glabra*	T24	IC_50_ 55 µM; 24 h	Inhibition of proliferation		[31]
			20, 40, 60, 80 µM; 2, 4, 6, 8, 12 h		↑ROS	
			60, 80 μM		↓GSH/GSSG	
Luteolin		T24	10, 25 µmol/L; 48 h	Induction of apoptosis	↑Cdc25c, Cdk2, p21^Waf1/Cip1^, p27^Kip1^, p53	[32]
		5637	25 µmol/L; 48 h		↓Cdk4, cyclin A, cyclin D1, p-S6, p-p70S6K, ROS	
N-butylidenephthalide (BP) from *Radix Angelica Sinensis*	*Radix Angelica Sinensis*	BFTC	60 μg/mL; 72 h	Induction of apoptosis	↑caspase -3, -9, E-cadherin ↓N-cadherin	[33]
OAPF	*Uncaria tomentosa*	T24, RT4	IC_50_ 164.13 ± 10.12 μg/mL, 137.23 ± 11.77 μg/mL; 48 h	Induction of cytotoxicity		[34]
OAPF 15′	*Uncaria tomentosa*	T24, RT4	IC_50_ 154.86 ± 16.61 μg/mL, 132.25 ± 25.33 μg/mL; 48 h	Induction of cytotoxicity		[34]
OAPF 45′	*Uncaria tomentosa*	T24, RT4	IC_50_ 182.83 ± 19.35 μg/mL, 153.00 ± 21.12 μg/mL; 48 h	Induction of cytotoxicity		[34]
OAPF 5′	*Uncaria tomentosa*	T24, RT4	IC_50_ 175.21 ± 35.04 μg/mL, 124.22 ± 17.84 μg/mL; 48 h	Induction of cytotoxicity		[34]
Salidroside	*Rhodiola rosea* L.	UMUC-3/pEGFP-LC3	25, 50 μg/mL; 8 h	Induction of autophagy	↑LC3-GFP puncta ↓p62	[35]
Saponin **1**	*Pittosporum angustifolium*	5637	IC_50_ 4.1 µM; 72 h	Inhibition of proliferation		[36]
Saponin **10**	*Pittosporum angustifolium*	5637	IC_50_ 2.4 µM; 72 h	Inhibition of proliferation		[36]
Saponin **2**	*Pittosporum angustifolium*	5637	IC_50_ 5.2 µM; 72 h	Inhibition of proliferation		[36]
Saponin **3**	*Pittosporum angustifolium*	5637	IC_50_ 2.1 µM; 72 h	Inhibition of proliferation		[36]
Saponin **4**	*Pittosporum angustifolium*	5637	IC_50_ 17.9 µM; 72 h	Inhibition of proliferation		[36]
Saponin compound **12**	*Allyl isothiocyanate*	ECV-304	IC_50_ 8.5 μg/mL; 72 h	Inhibition of proliferation		[37]
Saponin compound **13**	*Allyl isothiocyanate*	ECV-304	IC_50_ 14.3 μg/mL; 72 h	Inhibition of proliferation		[37]
Saponin compound **15**	*Allyl isothiocyanate*	ECV-304	IC_50_ 5.8 μg/mL; 72 h	Inhibition of proliferation		[37]
Sulforaphane	*Brassica oleracea var. italica*	UM-UC-3 cell bearing athymic mice	12 mg/kg; 5 weeks	Suppression of cancer cell growth	↑caspase 3, cytochrome c ↓survivin	[38]
Sulforaphane	*Brassica oleracea var. italica*	UMUC3	20 μM; 48 h	Induction of apoptosis	↓survivin, EGFR, HER2/neu, Ki67	[22]
		UMUC3 bearing female athymic mice	295 μmol/kg/day; 2 weeks	Suppression of cancer cell growth		
TA-NE-F4	*Salvia miltiorrhiza bunge*	T24	IC_50_ 13.48 ± 4.30 μg/mL; 24 h	Induction of cytotoxicity		[39]
Tea polyphenols (TP)		T24	EPI = 8.6 μM TP = 399.6 μM	Inhibition of autophagy and induction of apoptosis	↑LC3-II, JNK ↓p62, Bcl-2	[40]
		BIU87	EPI = 38.3 μM TP = 670.8 μM			
TLC-pure f61	*Cassia alata* L.	T24	25, 50, 100 µg/ mL IC_50_ 17.13 µg/mL 50 µg/mL; 12–24 h	Induction of cytotoxicity		[27]
				Induction of apoptosis-related morphology		
Triterpenoid saponin **1**	*Ardisia gigantifolia*	EJ	IC_50_ 3.4 ± 0.1 µM; 48 h	Induction of cytotoxicity		[41]
Triterpenoid saponin **2**	*Ardisia gigantifolia*	EJ	IC_50_ 4.0 ± 0.4 µM; 48 h	Induction of cytotoxicity		[41]
Triterpenoid saponin **4**	*Ardisia gigantifolia*	EJ	IC_50_ 2.0 ± 0.1 µM; 48 h	Induction of cytotoxicity		[41]
Triterpenoid saponin **5**	*Ardisia gigantifolia*	EJ	IC_50_ 3.1 ± 0.2 µM; 48 h	Induction of cytotoxicity		[41]
Yuanhuacine (YHL-14)	*Daphne genkwa Siebold et Zucc.*	T24T, UMUC3	2–16 μm; 24 h	Suppression of cancer cell growth		[42]
		T24T	2 μm; 12 h		↑p21, Sp-1, p38	
β-Eudesmol	*Murraya tetramera*	EJ	IC_50_ 31.93 µg/mL; 48 h	Inhibition of proliferation		[15]

Footer: EJ, human endometrial adenocarcinoma; UM-UC, human bladder transitional cell carcinoma; miR, microRNA; ↑, up-regulation; TIMP, metalloproteinase; JNK, Jun N-terminal kinase, stress-activated protein kinase; ERK, Extracellular signal-regulated kinase; Jun, Jun family; Akt, Protein kinase B (PKB); GSK, Glycogen Synthase Kinase; ↓, down-regulation; AR, adenosinereceptor; Lef, lymphoid enhancer factor; Tcf, T cell factor; myc, Myc family; MB, Mouse Bladder; TSGH, human gastric carcinoma cell line; H3K27me3,an epigenetic modification to the DNA packaging protein Histone H; pH3Ser10, H3 phosphorylation at serine 10; FABP4, fatty acid-binding protein 4; HBp17, Heparin-Binding Protein 17; RGS4, Regulator of G protein signaling 4; UMUC3, an epithelial-like cell that was isolated from the urinary bladder male of a patient and can be used in cancer research; EGFR, estimated glomerular filtration rate; HER, human epidermal growth factor receptor 2; Ki67, a nuclear antigen that is an excellent marker of active cell proliferation in the normal and tumor cell populations; TCC, transitional cell cancer; PTEN, phosphatase and tensin homolog; ROS, reactive oxygen species; Bax, BCL2-Associated X-Protein; Bcl, B-cell lymphoma; Bad, BCL2 associated agonist of cell death; Bid, BH3 Interacting Domain Death Agonist; Mcl, myeloid cell leukemia; SW620, the human colon cancer cell line; MCF-7, a breast cancer cell line; T24R2, a cisplatin-resistant derivative cell line of T24; HEK293, immortalized human embryonic kidney cells; BALB, bagg and albino; GSH, glutathione; GSSG, glutathione disulfide; Cdc25c, a dual specificity phosphatase essential for dephosphorylation; Cdk, Cyclin-dependent kinase; BFTC, Black Foot disease Transitional Carcinoma; EGFP, enhanced green fluorescent protein; LC3, Microtubule-associated protein 1A/1B-light chain 3; ECV-304, a spontaneously-transformed line derived from a Japanese human umbilical vein endothelial cells (HUVEC) culture; BIU87, Cellosaurus cell line.

**Table 2 pharmaceutics-15-01117-t002:** Single extracts inducing apoptosis.

Compound/Extract	Source	Experimental Model	Dose; Duration	Efficacy	Mechanism	Ref.
Acetone extract of leaves and branchlets	*Juniperus foetidissima*	EJ-138	=43.263.21, 44.274.25 and 26.172.96 μM	Induction of proliferation and cytotoxicity		[44]
		Caov-4	=25.723.13, 57.384.56 and 37.354.13 μM			
Black bryony aqueous extract	*Tamus communis* L.	T24	0.5, 1, 2 mg/mL; 48 h	Induction of apoptosis		[45]
Brazilian red propolis ethanolic extract		5637	50 μg/mL; 24 h	Induction of early apoptosis	↑Bcl-2, Bax, AIF, caspase-9, caspase-3, p53 ↓Bax/Bcl-2 ratio	[46]
			100 μg/mL; 24 h	Induction of early apoptosis, late apoptosis or death	↑Bax/Bcl-2 ratio ↓Bax, Bcl-2, caspase-3, caspase-8, caspase-9	
*Brucea javanica* (L.) Merr. Oil	*Brucea javanica* (L.) Merr.	T24	0.078, 0.156, 0.313, 0.625, 1.25, 2.5, 5 mg/mL; 48 h	Induction of apoptosis	↑caspase-3, -9 ↓NF-κB p65, COX-2	[47]
Butcher’s broom aqueous extract	*Ruscus aculeatus* L.	T24	0.5, 1, 2 mg/mL; 48 h	Induction of apoptosis		[45]
*Celastrus kusanoi* Hayata CHCl3 extracts from stems	*Celastrus kusanoi* Hayata	NTUB1	5, 10 μM; 24 h	Induction of cell cycle arrest and apoptosis	↑ROS	[48]
Dichloromethane Leaf extract	*Lepidium latifolium* L.	UM-UC-3	IC_50_ 133.8 μg/mL; 24 h	Induction of cytotoxicity		[16]
Ethanol extracts of peel (EECU)	*Citrus unshiu Marknovich*	T24	0, 100, 200, 400, 600, 800, 1000 μg/mL; 48 h	Induction of apoptosis	↑ROS, Caspase -8, -9, -3 ↓P13K, Akt, LY294002	[49]
Garlic extract	*Allium sativum* L.	T24 bearing BALB/c mice	20, 200, 1000 mg/kg; 43 days	Inhibition of proliferation		[50]
			1000 mg/kg; 43 days		↑AKAP12, RDX ↓RAB13	
Glycoalkaloidic extract	*Solanum lycocarpum*	RT4	5, 10 μg/mL; 24 h	Induction of apoptosis		[51]
Grape seed extract (GSE)		T24	50, 100 μg/mL; 24, 48 h	Induction of apoptosis	↑cleaved caspase-3, -9, cleaved-PARP ↓Mcl-1	[52]

		HTB9	25, 50, 100 μg/mL; 24, 48 h	Induction of apoptosis	↑cleaved caspase-3, -9, cleaved-PARP ↓Mcl-1	
Herbal extract	*Guizhi Fuling Wan*	BFTC 905, TSGH 8301	0, 0.5, 1 g/mL; 24 h	Induction of cell cycle and apoptosis	↑CHK2, p21 ↓CDK2-cyclin E, A, and A complexes	[53]
*Ipomoea obscura* (L.) *alcoholic extract*	*Ipomoea obscura* (L.)	Swiss albino mice	10 mg/kg; 4, 24, 48 h	Induction of uro- and nephrotoxicities	↑IFN-γ, IL-2 ↓TNF-α	[54]
Lemongrass essential oil	*Cymbopogon citratus* Stapf	MNU-treated female BALB/c mice	500 mg/kg; 5 weeks	Reduction in proliferation and apoptotic index		[55]
*Lycium barbarum polysaccharides* (LBP)	*Lycium barbarum*	BIU87	400, 800, 1200 μg/mL; 48 h	Inhibition of proliferation	↑P21, P27 ↓p-AKT	[56]
Methanol extract of the roots and solid-phase extraction (SPE) fractions	*Trillium govanianum*	MCF7, HepG2, A549, EJ138	=5, 7, 9, 5 μg/mL	Induction of Cytotoxicity		[57]
Mulberry water extract (MWE)	*Morus alba*	TSGH8301	25, 500, 750, 1000, 1500 μg/mL; 24, 48 h	Increase in apoptotic effect of paclitaxel	↑p-cdc2, p-cyclin B1, p-aurora A, p-plk1, PTEN, caspase-3 ↓EEA1	[58]
Pentacyclic oxindole alkaloids (POA) from stem bark and leaves of *Uncaria tomenotosa*	*Uncaria tomenotosa*	Human leukocytes	=33.80 to 736.23 μM (OAPFs) and 44.32 μM (CE)	Induction of genotoxicity and cytotoxicity		[59]
		T24	=181.68 to 267.05 μM (OAPFs) and =9.54 μM (CESII)			
		U-251-MG	=351.64 to 403.50 μM (OAPFs) and CESII being ineffective			
Polyphenolic fraction from *Punica granatum* L.	*Punica granatum* L.	T24	50 μg/mL; 48 h	Inhibition of oxidation and proliferation		[60]
Pomegranate fruit ethanol extract	*Punica granatum*	T24	50, 100 μg/mL; 24, 48, 72 h	Induction of apoptosis	↑pro-caspase-3, -8,-9, -12 Bax/Bcl-2 ratio, CHOP, Bip	[61]
Pomegranate fruit ethanol extract (PFE)	*Punicagranatum*	T24, J82, TSGH8301	50 μg/mL; 72 h	Inhibition of proliferation		[62]
		T24	50 μg/mL; 48 h		↑profilin 1 ↓PTEN, AKT, p-AKT, mTOR	
		T24	50 μg/mL; 48 h	Induction of apoptosis	↑Diablo ↓XIAP	
spray-dried acai pulp (AP)	*Euterpe oleraceae* Martius	male Swiss mice	standard diet containing 5% AP; 10 weeks	Inhibition of proliferation	↓p63, PCNA	[63]
		male Swiss mice	standard diet containing 5% AP; 3 weeks		↓DNA damage levels induced by H2O2	
*Tinospora cordifolia* alcoholic extract	*Tinospora cordifolia*	Swiss albino mice	200 mg/kg; 4, 24, 48 h	Induction of apoptosis	↑GSH, IFN-γ, IL-2 ↓TNF-α	[64]
Wild asparagus aqueous extract	*Asparagus acutifolius* L.	T24	0.5, 1, 2 mg/mL; 48 h	Induction of apoptosis		[45]

Footer: ↑, Up-regulation; Bcl-2, B-cell lymphoma 2; Bax, Bcl-2 Associated X-protein; AIF, Apoptosis-inducing factor; ↓, Down-regulation; NF-κB, Nuclear factor kappa-light-chain-enhancer of activated B cells; COX, Cyclooxygenase; ROS, reactive oxygen species; PI3K, Phosphoinositide 3-kinase; AKT, Protein kinase B; AKAP12, A-Kinase Anchoring Protein 12; RDX, Radixin; PARP, Poly ADP-ribose polymerase; Mcl-1, Myeloid cell leukemia 1; Cdk2, Cyclin-dependent kinase 2; IFN-γ, Interferon gamma; IL-2, Interleukin-2; TNF-α, Tumor necrosis factor α; PTEN, Phosphatase and tensin homolog; EEA1, Early Endosome Antigen 1; CHOP, CCAAT-enhancer-binding protein homologous protein; Bip, Binding immunoglobulin protein; PTEN, Phosphatase and tensin homolog; mTOR, mammalian target of rapamycin (mTOR); XIAP, X-linked inhibitor of apoptosis protein; PCNA, Proliferating cell nuclear antigen; GSH, Reduced glutathione.

**Table 3 pharmaceutics-15-01117-t003:** Mixture Extracts inducing Apoptosis.

Compound/Extract	Source	Experimental Model	Dose; Duration	Efficacy	Mechanism	Ref.
Green coffee bean and yerba mate extracts (GCBE and YME)	*Coffea arabica and* *llex paraguariensis*	Caco-2, A549, OE-33, T24, CCD-18Co	0.1, 1, 10 μg/mL; 2, 24 h	Inhibition of proliferation and induction of cytotoxicity	↑caspases-8 and -3 ↓NF-kB, COX-2/PGE2, iNOS/NO, Topoisomerase II	[65]
Herba and flower aqueous extracts QCSL	*Centaurea ragusina* L. *Astragalus propinquus* Schischkin *Sagittaria* *sagittifolia* L. *Polyporus umbellatus* *Poria cocos* *Paeonia lactiflora Pall* *Curcuma zedoaria* *Cinnamomum cassia* *Glycyrrhiza glabra* L. *Rehmannia glutinosa* *Smilax glabra Roxb.*	T24, A1235 BALB/c-nu nude T24 xenograft mice	flower AE 2 g/L, herba AE 1 g/L; 4, 24, 48, 72 h 100, 200, 400 mg/kg; once a day for 7 weeks	Induction of cytotoxicity and apoptosis	↓sulphydryl enzymes, phosphofructokinases, glycogen	[66]
				Inhibition of tumor growth and proliferation, Induction of apoptosis	↓β-catenin, survivin, c-myc, cyclin D1	[67]
			400 mg/kg	Increase in survival time		

Footer: ↑, Up-regulation; NF-κB, Nuclear factor kappa-light-chain-enhancer of activated B cells; COX, Cyclooxygenase; PGE2, Prostaglandin E2; iNOS, Inducible nitric oxide synthase; NO, Nitric oxide; AE, Aqueous extracts; ↓, Down-regulation; BALB, Bagg and albino.

**Table 6 pharmaceutics-15-01117-t006:** Natural Products sensitizing Drug Resistance.

Classification	Compound/Extract	Source	Cell Line/Animal Model	Dose; Duration	Efficacy	Mechanism	Ref.
polyphenols	Resveratrol (RES)	peanuts, mulberries, peel of grapes	pumc-91/ADM	10, 50, 100 µM; 48 h	Reversion of drug resistance	↑Topo-II ↓MRP1, LRP, GST, BCL-2	[81]

Footer: pumc-91/ADM, adriamycin (ADM)-resistant pumc-91 cells; ↑, up-regulation; Topo-II, topoisomerase II; ↓, down-regulation; MRP1, multi-drug resistance 1; LRP, lung resistance protein; GST, glutathione S-transferase; BCL-2, B-cell leukemia/lymphoma 2.

## Data Availability

Data available in a publicly accessible repository that does not issue DOIs.

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
