# Peer review of "Natural Products as New Approaches for Treating Bladder Cancer: From Traditional Medicine to Novel Drug Discovery"

_pharmaceutics, 2023, doi:10.3390/pharmaceutics15041117_

Round 1

Reviewer 1 Report

1.       Keywords must be arranged alphabetically.

2.       Authors are suggested to mention the chemical structures and their numbering of Chemical constituent or phytochemicals presented in the manuscript.

3.        Conclusion part must be rewrite in more effective way.

4.       Plant names mentioned in the manuscript should be representing in similar ways like abbreviation pattern to be followed with each botanical name.

Author Response

We appreciate reviewers and editors for giving us an opportunity to resubmit our manuscript. We earnestly responded to the raised comments point by point.

  1. Keywords must be arranged alphabetically.

(Response): Revised.

  1. Authors are suggested to mention the chemical structures and their numbering of Chemical constituent or phytochemicals presented in the manuscript.

(Response): Thanks. Added in the manuscript.

  1. Conclusion part must be rewrite in more effective way.

(Response): Revised.

  1. Plant names mentioned in the manuscript should be representing in similar ways like abbreviation pattern to be followed with each botanical name.

(Response): We mainly mentioned the plant names in scientific name, but if it was described as a biopharmaceutical name or so in the paper we reviewed, we followed it.

Again, we appreciate reviewers and editors for their kind and careful comments for improving the quality of our manuscript and also sincerely hope we address our responses well to the raised comments and our revised manuscript would be accepted for publication in your journal soon.

With kind regards,

Prof. Bonglee Kim, M.D, Ph.D.

-Associate Professor of Department of Pathology, College of Korean Medicine, Kyung Hee University, 26 Kyungheedae-ro, Dongdaemun-gu, Seoul, 02453, Republic of Korea

-Chair of Department of Cancer Preventive Material Development, Kyung Hee University

-Group leader of Korean Medicine-Based Drug Repositioning Cancer Research Center

Phone: +82-2-961-9217 (South Korea)

Reviewer 2 Report

1. Table titles are inappropriate and makes no sense. 

2. For example, Saponin 1 in the compound column of Table 1 is supposed to indicate the name of a specific compound, but it cannot be identified. Since there must be other compound names, the compound must be indicated by a compound name that can identify the compound. There are many other compounds besides Saponin 1 that are listed with similar compound names and should be corrected in the same way.

Author Response

We appreciate reviewers and editors for giving us an opportunity to resubmit our manuscript. We earnestly responded to the raised comments point by point.

  1. Table titles are inappropriate and makes no sense. 

For example, Saponin 1 in the compound column of Table 1 is supposed to indicate the name of a specific compound, but it cannot be identified. Since there must be other compound names, the compound must be indicated by a compound name that can identify the compound. There are many other compounds besides Saponin 1 that are listed with similar compound names and should be corrected in the same way.

 (Response): Thank you for your valuable comments. Table titles were all revised. About the saponin numbers, we followed the research article because the authors of the study didn’t reveal the name of their drugs.

Again, we appreciate reviewers and editors for their kind and careful comments for improving the quality of our manuscript and also sincerely hope we address our responses well to the raised comments and our revised manuscript would be accepted for publication in your journal soon.

With kind regards,

Prof. Bonglee Kim, M.D, Ph.D.

-Associate Professor of Department of Pathology, College of Korean Medicine, Kyung Hee University, 26 Kyungheedae-ro, Dongdaemun-gu, Seoul, 02453, Republic of Korea

-Chair of Department of Cancer Preventive Material Development, Kyung Hee University

-Group leader of Korean Medicine-Based Drug Repositioning Cancer Research Center

Phone: +82-2-961-9217 (South Korea)

Reviewer 3 Report

In the manuscript “Natural Products as New Approaches for Treating Bladder Cancer: From Traditional Medicine to Novel Drug Discovery”, Kang et al comprehensive investigates the latest progresses have been made in the treatment of Bladder Cancer based on natural products from the traditional medicine. The topic is interesting, and a good summary have been made in this manuscript. There are several comments to help the authors improve their manuscript.

1. There are several categorization criteria to identify the cellular grade, a comprehensive summary of current main classification standards and their clinical usage in the BC diagnosis is required.

2. Only papers published from 2016 to 2020 and papers written in English were reviewed. How about the latest research progresses of this area? It should be updated in time.

3. Surgical excision is the main treatment for BCs, and the Chemotherapy and immunotherapy combined with surgical excision or other approach are commonly used to treat bladder cancer. Does the natural products are also belonged to category of chemotherapy? What’s the potential advantage of the natural products in BCs treatment?

4. What’s the disadvantage of the of the natural products in cancer treatments. Is there any side effect of the natural products? A comprehensive discussion of the potential limitation of the natural products in the clinical will improve the readability remarkably.

5. Are there any natural products or its derivatives has been used in clinical, especially in BCs treatment?

6. The language should be improved remarkably, and ill format needs to be modified. For example, line 822, [27], [19]

Author Response

We appreciate reviewers and editors for giving us an opportunity to resubmit our manuscript. We earnestly responded to the raised comments point by point.

  1. There are several categorization criteria to identify the cellular grade, a comprehensive summary of current main classification standards and their clinical usage in the BC diagnosis is required.

(Response): Added corresponding contents dealing with BC diagnosis cellular grade, to the introduction part.

  1. Only papers published from 2016 to 2020 and papers written in English were reviewed. How about the latest research progresses of this area? It should be updated in time.

(Response): Sorry for the typo. We searched 2016-2023 studies but there was no study fulfill our criteria from 2021 to 2023.

  1. Surgical excision is the main treatment for BCs, and the Chemotherapy and immunotherapy combined with surgical excision or other approach are commonly used to treat bladder cancer. Does the natural products are also belonged to category of chemotherapy? What’s the potential advantage of the natural products in BCs treatment?

(Response): New agents have been approved for patients with BCG failure who faced radical cystectomy so far. And alternatives to perioperative chemotherapy have arisen to increase the likelihood of complete treatment delivery and successful oncological outcomes. Finally, improvements in molecular biology and our understanding of tumorigenesis open the era of personalized medicine in bladder cancer[1]. Furthermore, natural products are potent candidates for chemotherapy and could be used as supplements for cancer patients.

  1. What’s the disadvantage of the of the natural products in cancer treatments. Is there any side effect of the natural products? A comprehensive discussion of the potential limitation of the natural products in the clinical will improve the readability remarkably.

(Response): Certain natural products that interact with chemotherapy during chemotherapy should not be taken. Added to the discussion.

  1. Are there any natural products or its derivatives has been used in clinical, especially in BCs treatment?

(Response): For example, Scutellaria baicalensis, which contains baicalin as an ingredient. baicalin exert its anticancer activity by inducing apoptosis and cell death in bladder cancer cells.

  1. The language should be improved remarkably, and ill format needs to be modified. For example, line 822, [27], [19]

(Response): The manuscript is edited by native English speaker.

Again, we appreciate reviewers and editors for their kind and careful comments for improving the quality of our manuscript and also sincerely hope we address our responses well to the raised comments and our revised manuscript would be accepted for publication in your journal soon.

With kind regards,

Prof. Bonglee Kim, M.D, Ph.D.

-Associate Professor of Department of Pathology, College of Korean Medicine, Kyung Hee University, 26 Kyungheedae-ro, Dongdaemun-gu, Seoul, 02453, Republic of Korea

-Chair of Department of Cancer Preventive Material Development, Kyung Hee University

-Group leader of Korean Medicine-Based Drug Repositioning Cancer Research Center

Phone: +82-2-961-9217 (South Korea)

  1. Dobruch, J.; Oszczudłowski, M. Bladder Cancer: Current Challenges and Future Directions. Medicina 2021, 57, 749. https://doi.org/10.3390/medicina57080749

Round 2

Reviewer 2 Report

There is little information to be gained by listing compounds for which the compound name is not known.

At least, the molecular weight or other information that can distinguish each compound is necessary.

If they are not available, it would be more appropriate to comment on them in the text rather than listing them in the Table.

Author Response

Thank you for your valuable comments.

We added the molecular weight of the compounds in the text.

Reviewer 3 Report

Thanks for the author's reply. All of my comments have been well-solved. 

Author Response

Thank you very much for your valuable comments.